# Proton mediated spin state transition of cobalt heme analogs

Jianping Zhao[1,8], Qian Peng [2,8], Zijian Wang[2], Wei Xu [3], Hongyan Xiao [4], Qi Wu[1], Hao-Ling Sun[5], Fang Ma[5], Jiyong Zhao[6], Cheng-Jun Sun[6], Jianzhang Zhao[7] & Jianfeng Li [1]

The spin state transition from low spin to high spin upon substrate addition is one of the key steps in cytochrome P450 catalysis. External perturbations such as pH and hydrogen bonding can also trigger the spin state transition of hemes through deprotonated histidine (e.g. Cytochrome $c$). In this work, we report the isolated 2-methylimidazole Cobalt(II) [Co(TPP) (2-MeHIm)] and [Co(TTP)(2-MeHIm)], and the corresponding 2-methylimidazolate derivatives where the N−H proton of axial 2-MeHIm is removed. Interestingly, various spectroscopies including EPR and XAFS determine a high-spin state ($S = 3/2$) for the imidazolate derivatives, in contrast to the low-spin state ($S = 1/2$) of all known imidazole analogs. DFT assisted stereoelectronic investigations are applied to understand the metal-ligand interactions, which suggest that the dramatically displaced metal center allowing a promotion $e_g(d_\pi) \rightarrow b_{1g}(d_{x^2-y^2})$ is crucial for the occurrence of the spin state transition.

[1] College of Materials Science and Opto-electronic Technology, CAS Center for Excellence in Topological Quantum Computation, & Center of Materials Science and Optoelectronics Engineering, University of Chinese Academy of Sciences, Yanqi Lake, Huairou District, 101408 Beijing, China. [2] State Key Laboratory and Institute of Elemento-Organic Chemistry, College of Chemistry, Nankai University, 300071 Tianjin, China. [3] Institute of High Energy Physics & University of Chinese Academy of Sciences, Chinese Academy of Sciences, 100049 Beijing, China. [4] Key Laboratory of Photochemical Conversion and Optoelectronic Materials, Technical Institute of Physics and Chemistry, Chinese Academy of Sciences, 100190 Beijing, China. [5] Department of Chemistry and Beijing Key Laboratory of Energy Conversion and Storage Materials, Beijing Normal University, 100875 Beijing, China. [6] Advanced Photon Source, Argonne National Laboratory, Argonne, IL 60439, USA. [7] State Key Laboratory of Fine Chemicals, School of Chemical Engineering, Dalian University of Technology, West Campus, 2 Ling-Gong Road, 116024 Dalian, China. [8] These authors contributed equally: Jianping Zhao, Qian Peng. Dedicated to the 40th anniversary of the University of Chinese Academy of Sciences and the 100th anniversary of Nankai University. Correspondence and requests for materials should be addressed to J.L. (email: jfli@ucas.ac.cn)

Spin state transition of hemes that is usually accompanied by metal displacement and conformational changes widely exists in biochemical processes, e.g., oxygenation of myoglobin (5c, HS (5c = five-coordinate, HS = high-spin) towards 6c, LS (6c = six-coordinate, LS = low-spin))[1,2], and the starting step of Cytochrome p450 catalysis (6c, LS towards 5c, HS)[3,4], both involved with ligand (un)binding. Besides this, spin state transition can also be controlled by (weak) external perturbations such as pH and hydrogen bonding[5]. The proximal His18 of Cytochrome $c$ (or microperoxidases) can be deprotonated at high pH (> 11) to trigger the pH-dependent spin state transition (HS towards LS)[5–7]. Although the alkaline transition has been studied for decades, questions remain regarding the nature of the trigger group that on deprotonation initiates the electronic and conformational change in the native molecule[6]. Similarly, the hydrogen bonding between Asp and proximal histidine in oxygen activating heme enzymes (i.e., peroxidases) contributes to negative imidazolate character to promote heterolytic cleavage of the O−O bond to form a ferryl species[8–10]. The origin of the process and electronic structural changes whereby enzymes reorganize their active site through external perturbations is still poorly understood. Porphyrin system has been used to investigate the impact of external perturbations for various advantages, including the isolable active site where the spin state transition happens[11]. Hydrogen bonding with axial chloride has been found to control the spin state of iron(III) octaethyltetraarylporphyrin chloride to switch between high ($S = 5/2$) and intermediate spin ($S = 3/2$)[12]. To imitate the deprotonated histidine or the extreme case of strong hydrogen bonding, imidazolate, which is produced by removing the hydrogen of imidazole to give stronger σ and π donation, has been exploited[13,14]. Scheidt and coworkers[15] have applied this idea in the studies of iron(II) porphyrinates where Mössbauer characterizations have revealed two different HS configurations, a complete spin state transition however is not seen.

In this work, we report the spin state transitions of synthetic metalloporphyrins, which are achieved through imidazole(ate) ligands that directly interact with the metal centers. The unexpected high-spin state of imidazolate cobalt(II) porphyrinates, as well as the incomplete spin state transition of iron(II) analogs[15], pose questions on the stronger ligand nature of imidazolate than imidazole.

## Results

**Single crystal structures.** Cobalt and iron hemes are analogous in many aspects, including the oxygen bonding ability[16–18]. Recently, we have reported the [Co(TpivPP)(R-Im)(O₂)] (R-Im: 1-EtIm or 2-MeHIm; TpivPP = α, α, α, α-tetrakis(o-pivalamidophenyl)porphyrinato) complexes using imidazole cobalt(II) porphyrinates as the starting material, which are all five-coordinate due to the destabilization of the six-coordinate compound by the singly populated $d_{z^2}$ orbital[19]. Remarkably all the imidazole cobalt(II) porphyrinates are low-spin state (3d⁷, $S = 1/2$) regardless of the steric hindrance of the axial ligands[20]. This is contrasted to iron(II) analogs, the use of hindered imidazole is necessary to prepare the five-coordinate iron(II) porphyrinates, all of which known so far are high-spin (3d⁶, $S = 2$)[21].

Here, we report the first examples of imidazolate ligated Co(II) porphyrinates [K(222)][Co(TPP)(2-MeIm⁻)] (222 = 4,7,13,16,21,24-hexaoxa-1,10-diazabicyclo [8.8.8] hexacosane) and [K(222)][Co(TTP) (2-MeIm⁻)] (TPP = tetraphenylporphyrin, TTP = tetratolylporphyrin), which are prepared by the reaction between [Co(TPP)] (or [Co(TTP)]) and [K(222)][(2-MeIm⁻)] (cryptand 222 is used to stabilize the K⁺ cation) (Supplementary Figs. 1–4). Two corresponding imidazole derivatives [Co(TPP)(2-MeHIm)] and [Co(TTP)(2-MeHIm)] are also isolated for comparison (Supplementary Table 1). The labeled ORTEP diagrams of the TPP and TTP derivatives are given in Fig. 1 and Supplementary Figs. 5 and 6. Quantitative information that shows the displacements of each atom from the 24-atom mean plane and the orientations of the axial ligands are available in Supplementary Fig. 7. "Shoestring" diagrams illustrating the core conformation and cobalt displacement from the N₄ mean plane are given in Supplementary Fig. 8. It is seen that [Co(TPP)(2-MeHIm)] shows the most distorted porphyrin core with saddled conformation among the four new structures, which can be attributed to the strong steric repulsion between imidazole and Nₚ−Co−N_Im planes evidenced by the smallest φ angle (8.5°) between them (Supplementary Fig. 7). The key structural parameters of all known imidazole(ate) ligated cobalt(II) porphyrinates are given in Table 1. Also given are the parameters of three pairs of 2-methylimidazole(ate) ligated iron(II) analogs. It is seen all the cobalt(II) complexes with neutral imidazole ligands including the two new structures have (Co–Nₚ)av

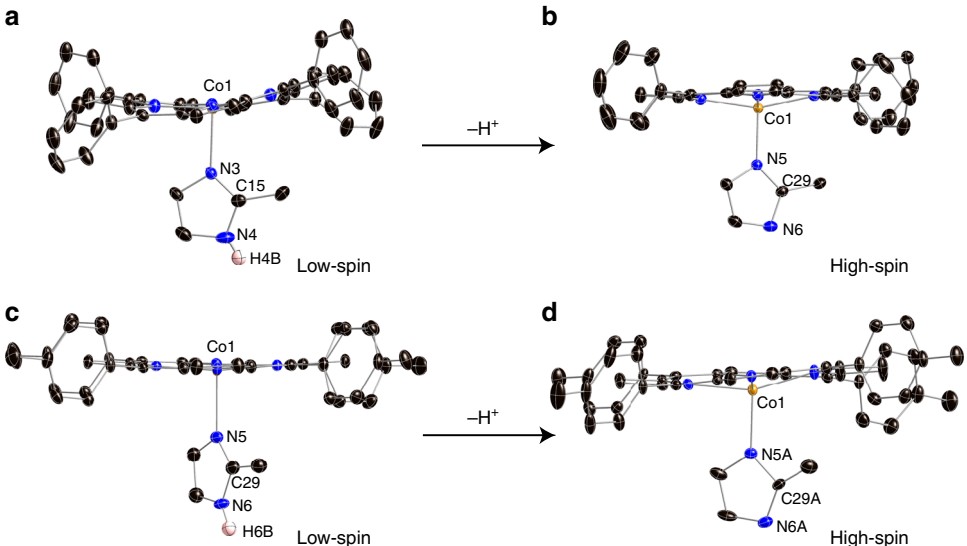

**Fig. 1** ORTEP diagrams. **a** [Co(TPP)(2-MeHIm)]. **b** [Co(TPP)(2-MeIm⁻)]⁻. **c** [Co(TTP)(2-MeHIm)] (one of the two orientations is shown for the axial imidazole). **d** [Co(TTP)(2-MeIm⁻)]⁻. Thermal ellipsoids of all atoms are contoured at the 50% probability level. Hydrogen atoms, [K(222)]⁺ and solvent molecules are not shown for clarity

**Table 1 Selected structural parameters of imidazole(ate) ligated cobalt/iron porphyrinates**

| Complex | Method | $\Delta_{24}$[a,b] | $\Delta_4$[a,b] | $(M-N_p)_{av}$[a,c] | $M-N_{Im}$[a,d] | $M-N_{Im}-C_{Im}(2)$[e,f] | $M-N_{Im}-C_{Im}(4)$[e,g] | $\varphi$[e,h] | $\tau$[e,i] | Refs. [j] |
|---|---|---|---|---|---|---|---|---|---|---|
| Cobalt(II) complexes | | | | | | | | | | |
| [Co(TPP)(2-MeIm⁻)]⁻ | SC | 0.53 | 0.43 | 2.079(7) | 2.0329(16) | 132.43(13) | 123.77(13) | 19.9 | 5.4 | tw |
| [Co(TPP)(2-MeIm⁻)]⁻ | DFT | 0.44 | 0.45 | 2.095(11) | 2.020 | 131.22 | 124.64 | 8.8 | 6.7 | tw |
| [Co(TTP)(2-MeIm⁻)]⁻ | SC | 0.47 | 0.43 | 2.077(11) | 2.035(4) | 134.2(4) | 122.3(5) | 23.3 | 6.0 | tw |
| | | | | | 2.034(9) | 130.8(9) | 127.9(8) | 23.8 | 13.8 | tw |
| [Co(TPP)(2-MeHIm)] | SC | 0.26 | 0.17 | 1.975(4) | 2.177(3) | 132.2(2) | 122.4(2) | 8.5 | 7.4 | tw |
| [Co(TPP)(2-MeHIm)] | DFT | 0.09 | 0.10 | 1.978(3) | 2.240 | 131.13 | 121.74 | 26.6 | 4.2 | tw |
| [Co(TTP)(2-MeHIm)] | SC | 0.19 | 0.15 | 1.988(3) | 2.1882(17) | 135.00(16) | 119.85(14) | 35.7 | 6.2 | tw |
| [Co(TpivPP)(2-MeHIm)] | SC | 0.15 | 0.14 | 1.979(3) | 2.145(3) | 132.0(3) | 123.1(3) | 21.6 | 6.5 | 20 |
| [Co(TPP)(1,2-Me₂Im)] | SC | 0.18 | 0.15 | 1.985(2) | 2.216(2) | 132.6(2) | 122.5(2) | 20.0 | 5.2 | 22 |
| [Co(TPP)(1-MeIm)] | SC | 0.14 | 0.13 | 1.977(3) | 2.157(3) | 127.8(3) | 126.4(3) | 4.1 | 7.2 | 23 |
| [Co(OEP)(1-MeIm)] | SC | 0.16 | 0.13 | 1.96(1) | 2.15(1) | 127(1) | 126(1) | 9.7 | 1.7 | 24 |
| [Co(OC₃OP)(1-MeIm)] | SC | 0.13 | 0.12 | 1.985(6) | 2.132(3) | 129.2(3) | 126.6(3) | 15.6 | 2.6 | 25 |
| Iron(II) complexes | | | | | | | | | | |
| [Fe(TPP)(2-MeIm⁻)]⁻ | SC | 0.66 | 0.56 | 2.118(13) | 1.999(5) | 129.6(3) | 126.7(3) | 23.4 | 9.8 | 15 |
| | | | | | 2.114(5) | 133.6(4) | 121.9(4) | 21.6 | 6.5 | 15 |
| [Fe(TPP)(2-MeIm⁻)]⁻ | SC | 0.61 | 0.55 | 2.113(11) | 2.0739(13) | 132.48(10) | 123.58(10) | 4.5 | 6.2 | 26 |
| [Fe(OEP)(2-MeIm⁻)]⁻ | SC | 0.65 | 0.56 | 2.113(4) | 2.069(2) | 136.6(2) | 120.0(2) | 37.4 | 3.6 | 15 |
| [Fe(TpivPP)(2-MeIm⁻)]⁻ | SC | 0.65 | 0.52 | 2.106(20) | 2.002(15) | NA[k] | NA[k] | 14.7 | 5.1 | 27 |
| [Fe(TPP)(2-MeHIm)] | SC | 0.38 | 0.32 | 2.073(9) | 2.127(3) | 131.3(3) | 122.9(2) | 24.0 | 8.3 | 28 |
| [Fe(OEP)(2-MeHIm)] | SC | 0.46 | 0.34 | 2.077(7) | 2.135(3) | 131.3(3) | 122.4(3) | 19.5 | 6.9 | 29 |
| [Fe(TpivPP)(2-MeHIm)] | SC | 0.38 | 0.35 | 2.070(6) | 2.113(3) | 128.5(2) | 125.7(3) | 23.3 | 8.3 | 30 |

[a]Values in angstroms
[b]Displacement of metal atom from the 24-atom ($\Delta_{24}$) or the four pyrrole nitrogen atoms ($\Delta_4$) mean plane. The positive numbers indicate a displacement towards the axial ligand
[c]Average distance between the metal and porphyrin nitrogen atoms
[d]Distance between the metal and the axial nitrogen atom
[e]Angle values in degrees
[f]$M-N_{Im}-C_{Im}$ angle with $C_{Im}$ being the 2-carbon of the ligand ring, sometimes methyl substituted
[g]$M-N_{Im}-C_{Im}$ angle with $C_{Im}$ being the 4-carbon of the ligand ring
[h]Dihedral angle between the ligand plane and the plane of the closest $N_p-M-N_{Im}$ (illustrated in Supplementary Fig. 7)
[i]The tilt of the $M-N_{Im}$ vector off the normal to the 24-atom mean plane
[j]tw this work
[k]Value not available

distances ≤ 2.0 Å, which is consistent with the low-spin Co(II) and small metal out of plane displacements ($\Delta_{24}$ and $\Delta_4$ ≤ 0.26 Å). This is contrasted to the two imidazolate derivatives that show longer (Co–$N_p$)av distances (≥ 2.0 Å) and unusually large metal out of plane displacements (≥ 0.43 Å), which are characteristic features of high-spin complexes[11]. Moreover, the two imidazolate ligands show shorter axial bonds (≤ 2.1 Å) than the imidazole (≥ 2.1 Å), indicating stronger axial bonding. Interestingly, similar patterns are also seen in the structures of iron(II) analogs that the imidazolate ligands always induce longer (M–$N_p$)av, larger metal out of plane displacements and shorter axial bonds (Table 1)[15,20,22–30]. In the study of imidazole(ate) ligated iron (II) porphyrinates, Scheidt and coworkers[15] have assigned two high-spin ground states $(d_{xy})^2(d_{xz}, d_{yz})^2(d_{z^2})^1(d_{x^2-y^2})^1$ and $(d_{xz}, d_{yz})^3(d_{xy})^1(d_{z^2})^1(d_{x^2-y^2})^1$ for 2-MeIm⁻ and 2-MeHIm derivatives, respectively. The authors suggested that the different electronic configurations caused the varying degree of electrostatic repulsion between the in-plane orbital of iron(II) and the negative charge of pyrrole nitrogen, which results in different structural features, including the iron displacements[15].

**Electron paramagnetic resonance**. The dramatically different structural parameters between imidazole and imidazolate species indicate different spin states of the Co(II) centers. To confirm this, electron paramagnetic resonance (EPR) experiments have been conducted on the four new complexes. The experimental and simulated spectra of crystalline [Co(TPP)(2-MeHIm)] and [Co(TPP)(2-MeIm⁻)]⁻ are given in Fig. 2. Multitemperature measurements on crystalline samples, solution samples with different equivalents of ligands, as well as those of TTP derivatives are available in Supplementary Figs. 9–16. As can be seen, the two

imidazole derivatives ([Co(TPP)(2-MeHIm)] and [Co(TTP)(2-MeHIm)]) show consistent axial symmetric spectra with $g_\perp$ = 2.3, $g_{//}$ = 2.0 ($A_{//}^{Co}$ = 79.2 G), which are typical for a five-coordinate low-spin Co(II)[31–35], in accordance with the single crystal structural features (e.g., small metal displacements and shorter (Co–$N_p$)av distances). In contrast, the two imidazolate derivatives ([Co(TPP)(2-MeIm⁻)]⁻ and [Co(TTP)(2-MeIm⁻)]⁻) show characteristic resonances at 6.0, 4.0, and 2.0 ($A_{//}^{Co}$ = 82.0 G) and 5.4, 3.9, 2.0, respectively, (Supplementary Figs. 15b and 16b), which corresponds to a high-spin Co(II) ($S$ = 3/2)[36–38], in agreement with the dramatically different structural parameters from the low-spin counterparts. The zero value of $E/D$ yielded by simulations confirmed the axial system where the | ± 3/2 > Kramers doublet is the excited state and resonances at 4.0 (3.9) and 2.0 come from the ground | ± 1/2 > doublet[39]. Notably, signals at ~2.3, which appear weak in solid while strong in solution samples, are observed in the spectra of imidazolate derivatives (the asterisk in Fig. 2b, Supplementary Figs. 9, 10, 13, and 14). To understand this, reactions of [Co(TPP)] (or [Co(TTP)]) with different equivalents of [K(222)(2-MeIm⁻)] in PhCl (or THF) were monitored and the spectra are given in Supplementary Fig. 13 (Supplementary Fig. 14). It is seen when 1 eq. of 2-MeIm⁻ was added to the [Co(TPP)] solution, resonances belonging to HS species (4.9–5.5 and 3.6–3.8) and a strong signal at ~2.3 became available immediately. Further addition of 2-MeIm⁻ (3, 5, and 7 eq.) has led to the increase of HS resonances, which suggests isolable [Co(TPP)(2-MeIm⁻)]⁻ product was generated gradually. In contrast, the signal at ~2.3, though decreasing relatively, was apparent even at the saturated solution (7 eq.). Hence, an intermediate of [Co(TPP)⋯(2-MeIm⁻)]⁻ with weakly bonding axial ligand, which is generated once 2-MeIm⁻ is added, is proposed to exist in the solution. Such a weak Co⋯(2-MeIm⁻) interaction

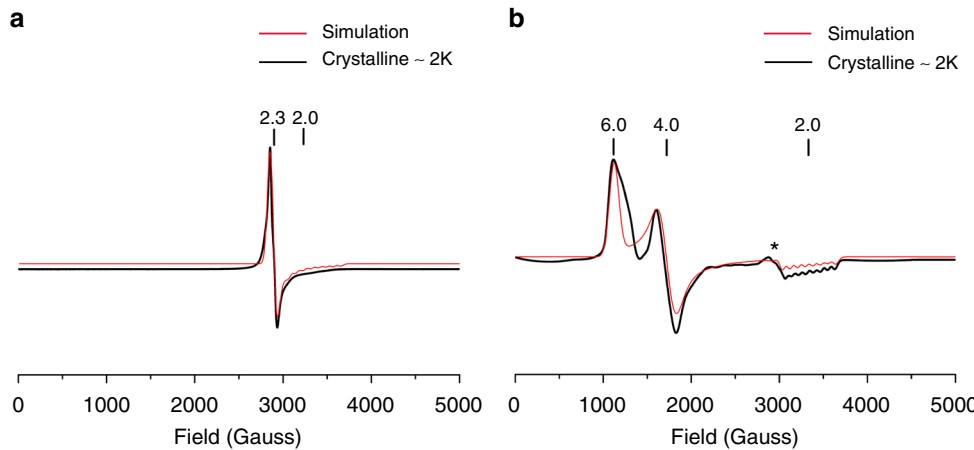

**Fig. 2** Experimental and simulated X-band EPR spectra. **a** [Co(TPP)(2-MeHIm)] and **b** [Co(TPP)(2-MeIm⁻)]⁻. The asterisk represents trace amount of [Co(TPP)••••(2-MeIm⁻)]⁻ intermediate

does not draw the metal out of porphyrin plane as much as the axial bond of isolable [Co(TPP)(2-MeIm⁻)]⁻ crystal; it is however similar to the longer axial bond of [Co(TPP)(2-MeHIm)], which shows low-spin signal at ~2.3 (*vide infra*). The intermediate also explains the weak signals observed in the crystalline samples since it can accompany the isolated crystals through the mother liquor as reported in other cases[40], and/or be generated during grinding processes (Supplementary Fig. 9).

**Magnetic susceptibility.** Additional evidence for the spin state determination of the four complexes comes from temperature dependent $(2 \rightleftarrows 30\,K)$ magnetic susceptibility measurements, which are given in Fig. 3. As can be seen, the product of the molar susceptibility $(\chi_M)$ and temperature (T) of the two imidazolate complexes are 2.03 and 1.93 cm³ K/mol, close to that expected for the HS state (1.88 cm³ K/mol). This is contrasted to the two imidazole derivatives, which show $\chi_M T$ at 0.49 and 0.54 cm³ K/mol, close to that expected for the LS state (0.38 cm³ K/mol). Hence, the magnetic susceptibility measurements are well consistent with the EPR results, both of which confirm the high- and low-spin states for imidazolate and imidazole derivatives, respectively.

**X-ray absorption spectroscopy.** X-ray absorption spectroscopy (XAS) studies have been conducted on [Co(TPP)(2-MeHIm)] and [Co(TPP)(2-MeIm⁻)]⁻ to give more insights into the electronic structures. The pre-edge features of Co $K$-edge X-ray absorption near edge structure (XANES), which originates from the $1s \rightarrow 3d$ states hybridized with p states of ligands (e.g., nitrogen) are given in Fig. 4a, b (black traces)[41]. Both species show identical pre-edge transition and the first inflection points at 7708.8 and 7719.3 eV (Supplementary Fig. 17), which are in accordance with the reported values for Co(II) complexes[42]. A shoulder feature along the rising edge at 7714.3 eV, which corresponds to the 1s to 4p + LMCT shakedown transition is observed for [Co(TPP)(2-MeHIm)][43,44]. Full multiple scattering theory (FMST) simulations are performed to interpret the XANES and given in Fig. 4a, b (red traces). The less sharp pre-edge of experimental spectra is due to the convoluted resolution[45]. The projection, integration (from −4 to 2 eV) and full width at half maximum (FWHM) of the unoccupied states of $t_{2g}$ and $e_g$ orbitals, which have been calibrated from the experimental data are given in Supplementary Figs. 18 and 19 and Supplementary Table 2[46,47]. Also given are the ratio of two unoccupied

states of $t_{2g}$ and $e_g$, which are calculated to be ~1.8 and ~1.4 for [Co(TPP)(2-MeHIm⁻)]⁻ and [Co(TPP)(2-MeHIm)], respectively. Since the HS state are expected to have more unoccupied $t_{2g}$ but less unoccupied $e_g$ to give a larger ratio value (Fig. 5), the XANES is parallel to the EPR, magnetic susceptibility, and single crystal characterizations, which suggest HS for the imidazolate derivatives. Fitting of EXAFS (Extended X-ray Absorption Fine Structure) spectra can be used to obtain geometric parameters of the first shell coordination around the Cobalt atom. The fitted EXAFS oscillations and the Fourier transforms are available in Fig. 4c, d. The fitting results are consistent with the crystal structures and given in Supplementary Table 3 for compariosn. Time-dependent density functional theory (TDDFT) simulations are also performed to interpret the XANES, which however underestimates the deep empty states as can be seen in Supplementary Figs. 20 and 21.

**Discussion**

As has been seen, the removal of N−H proton of 2-MeHIm has led to dramatic changes in both geometric and electronic structures of Co(II) porphyrinates. The changes appear unexpected because the imidazolate has been accepted as a strong ligand, which would induce LS species, e.g., in the alkaline transition[5–7]. Thus, we have conducted DFT calculations to investigate the products' electronic configurations (Supplementary Figs. 22–25 and Supplementary Data 1). M06/6-31G(d)/TZVP level of theory has predicted 2.6 and 5.7 kcal/mol more stable LS [Co(TPP)(2-MeHIm)] and HS [Co(TPP)(2-MeIm⁻)]⁻, respectively, in agreement with the experimental results. Qualitative diagrams showing the d-orbital energy ordering are illustrated in Fig. 5. The MOs of low and high-spin [Co(TPP)(2-MeIm⁻)]⁻ are compared in Fig. 5b. The dramatically lowered $d_{xz}$, $d_{yz}$, $d_{z^2}$ and $d_{x^2-y^2}$ orbitals of HS state, which is consistent with the short axial bond distance and large metal displacement are obvious (Table 1). Close examinations also found the anti-bonding interaction between the imidazolate π orbital and the $\beta$ orbital of $d_{yz}$ in LS state, which is 0.364 eV higher in energy than the $\alpha$ orbital of $d_{x^2-y^2}$ in HS state (gray arrow). These electronic features are in agreement with the experimentally observed HS state of [Co(TPP)(2-MeIm⁻)]⁻. The electronic structures of LS [Co(TPP)(2-MeHIm)] are given in Fig. 5a. The higher $d_{x^2-y^2}$ and the larger energy differences between $d_{x^2-y^2}$ and $d_{xz}/d_{yz}$ are seen, which is consistent with the small metal displacements of LS cobalt(II) (0.17 and 0.26 Å). Moreover, the near-degenerate $d_{xz}$ and $d_{yz}$

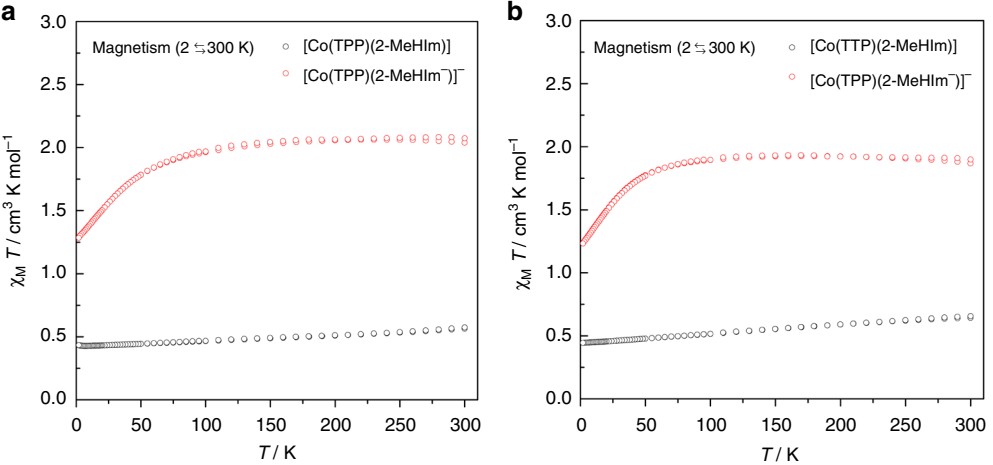

**Fig. 3** $\chi_M T$ vs. T in an external magnetic field of 1000 Gauss. **a** [Co(TPP)(2-MeHIm)] (black circles), [Co(TPP)(2-MeIm$^-$)]$^-$ (red circles). **b** [Co(TTP)(2-MeHIm)] (black circles), [Co(TTP)(2-MeIm$^-$)]$^-$ (red circles)

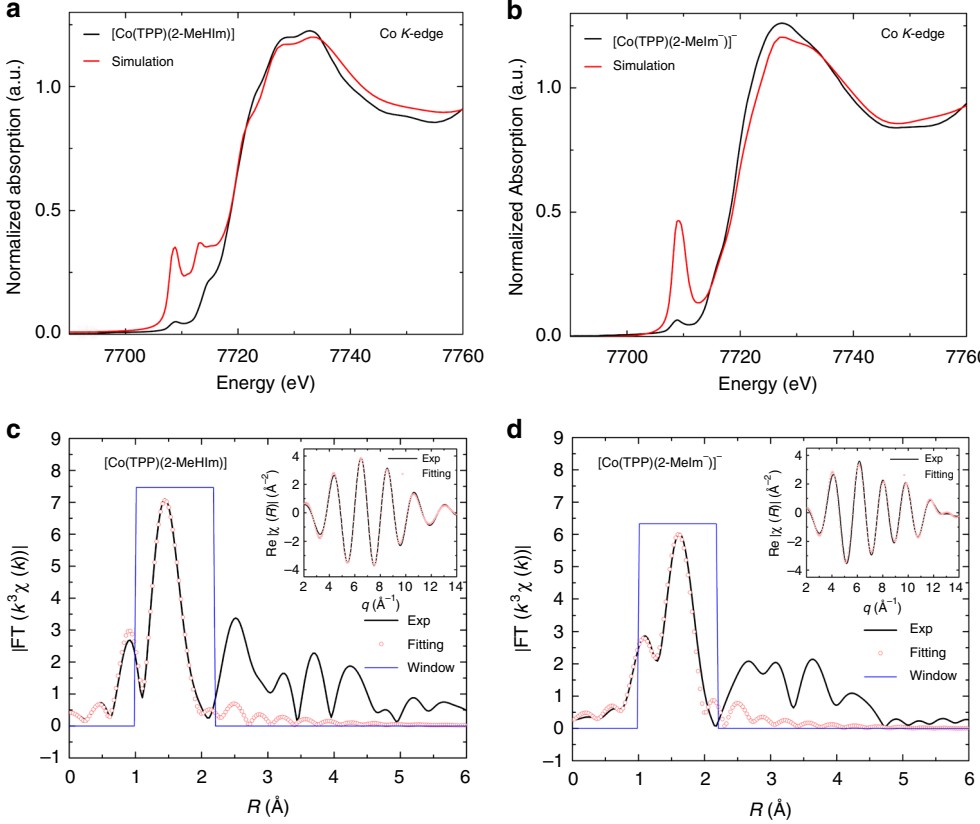

**Fig. 4** Comparisons of experimental and theoretical Co K-edge XANES and EXAFS. **a** K-edge XANES of [Co(TPP)(2-MeHIm)] (black line: Experiment, red line: Simulation). **b** K-edge XANES of [Co(TPP)(2-MeIm$^-$)]$^-$ (black line: Experiment, red line: Simulation). **c** EXAFS of [Co(TPP)(2-MeHIm)] (black line: Experiment, red circles: Fitting). **d** [Co(TPP)(2-MeIm$^-$)]$^-$ (black line: Experiment, red circles: Fitting)

orbitals, which is in agreement with the axial EPR resonances and longer axial distance, suggest the weaker imidazole ligation.

The spin state transition happened only to cobalt(II) porphyrinates in contrast to invariable high-spin states of 2-methylimidazole(ate) iron(II) analogs[15], although both pairs of counterparts are mediated by N−H proton of imidazole. Stereoelectronic analysis on interplays between imidazole(ate) and metal centers would reveal the differences between the two systems. It is suggested that 3d$^6$ iron(II), one electron less than 3d$^7$ Co(II), is in favor of stronger σ donation of imidazole(ate)

through d$_{z^2}$, which would draw the metal more out of the porphyrin plane, lower the d$_{x^2-y^2}$ orbital and make the HS states accessible. It is important to note that the steric hindrance is not required here because five-coordinate iron(II) porphyrinates with non-hindered imidazole (e.g., 1-MeIm) were also reported to be HS, which suggests the large iron displacement is induced mainly by strong axial bonding[48,49]. Interestingly, the switchable spin states of Co(II) analogs suggest 3d$^7$ Co(II), which always has one electron on d$_{z^2}$, has weaker tendency to the σ donation of axial ligand than 3d$^6$ iron(II). For the imidazole ligand, the relatively

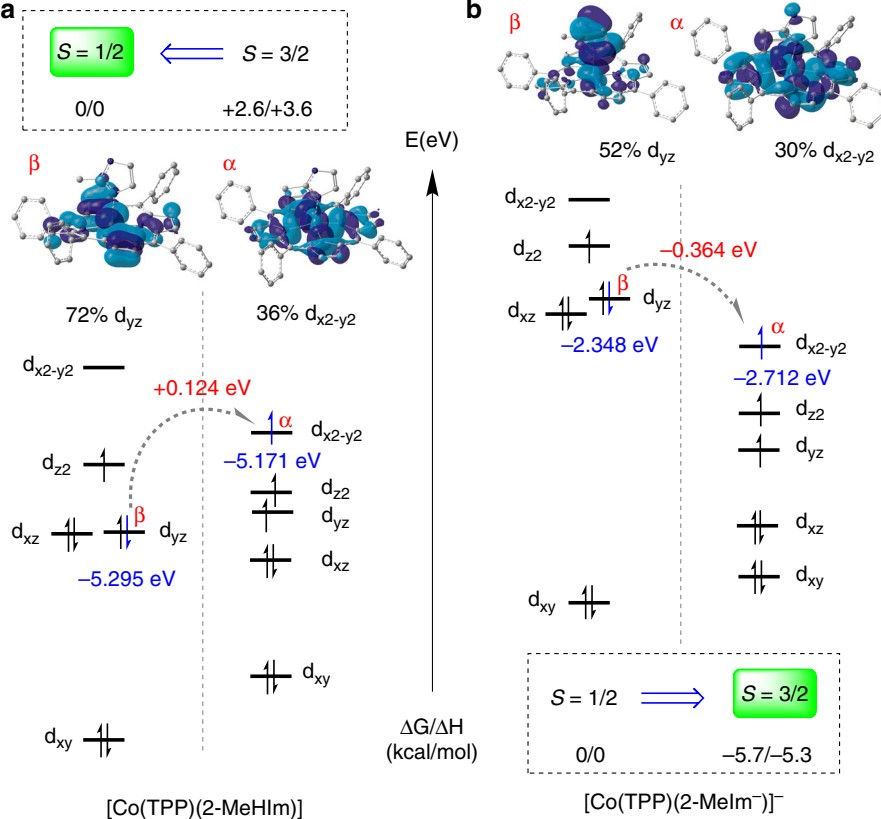

**Fig. 5** Qualitative diagrams showing the d-orbital energy ordering. **a** [Co(TPP)(2-MeHIm)] and **b** [Co(TPP)(2-MeHIm⁻)]⁻; iso-value = 0.02

weak axial bonding only induced small metal displacements (e.g., $\Delta_4$: 0.12–0.17 of [Co(Porph)(Im)] (Porph = Porphyrin) vs. 0.32–0.35 of [Fe(Porph)(2-MeHIm)] and 0.52–0.56 Å of [Fe(Porph)(2-MeIm⁻)]⁻, Table 1), so that the $d_{x^2-y^2}$ orbital is not drawn lower enough for the access of HS. This is true for both hindered and non-hindered imidazole as seen in Table 1, consistent with the LS states of all known five-coordinate [Co(Porph)(Im)] complexes. In contrast, the 2-methylimidazolate, a stronger ligand, has induced much shorter axial bonds and dramatically displaced Co(II), which drew the $d_{x^2-y^2}$ lower enough and made the $[(d_{xy})^2(d_\pi)^3(d_{z^2})^1(d_{x^2-y^2})^1]$ ($^4E_g$) configuration to be accessible.

Strategies to obtain a high-spin cobalt(II) heme complex have been proposed for many years. As early as 1983, Scheidt and Gouterman[50] predicted this to be achieved in five coordination with the metal out of plane, as this lowers the $d_{x^2-y^2}$ energy ($e_g(d_\pi) \rightarrow b_{1g}(d_{x^2-y^2})$). Later, DiMagno and coworkers[51] reported extremely electron-deficient β-octafluoro-*meso*-tetraarylporphyrins F$_{28}$TPP, which reduced porphyrin ligand field and stabilized $d_{x^2-y^2}$ orbital. Nevertheless, a HS product in solid state has never been isolated and characterized.

In summary, comprehensive spectroscopic characterizations on four isolated cobalt(II) heme complexes demonstrate the removal of N−H proton of axial 2-MeHIm has changed the metal centers from LS to HS, thus mimicking spin state transition of heme systems. The single crystal data highly promote stereoelectronic studies on the mechanisms of spin state transition, which reveal different response of iron(II) and cobalt(II) to the axial imidazole(ate) ligands and underline the stronger ligand field of imidazolate. The work also provides quantitative values for the metal displacements of hemes (e.g., $\Delta_{24}$ and $\Delta_4 \geq 0.2$ Å,

Table 1), which usually accompany with spin state transitions (and/or d-orbital reconstructions) that are physiologically important and can be triggered by charge changes of proximal ligands. To the best of our knowledge, this is the first examples of synthetic metalloporphyrins that can switch spin states through one proton of proximal ligands.

## Methods

**General procedure.** All reactions and manipulations were carried out under argon using a double-manifold vacuum line, Schlenkware, and cannula techniques unless otherwise noted. Tetrahydrofuran, chlorobenzene, and hexanes were dried and degassed by standard techniques. General considerations on the measurements and experiments, as well as some experimental details are described in the Supplementary Information.

**X-ray structure determinations.** Single crystal experiments were carried out on a BRUKER D8 QUEST system with graphite-monochromated Mo Kα radiation ($\lambda = 0.71073$ Å). The crystal samples were placed in inert oil, mounted on a glass fiber attached to a brass mounting pin, and transferred to the cold dinitrogen gas steam (100 K). Crystal data were collected and integrated using a Bruker Apex II system. The structures were solved by direct method (SHELXS-2014) and refined against $F^2$ using SHELXL-2014[52].

**EPR measurements and simulations.** EPR were carried out on a Bruker EMX plus 10/12 CW X-band EPR spectrometer, equipped with High-Q cylindrical cavity and Oxford ESR910 continuous flow liquid helium cryostat. The EPR spectra were simulated with the EasySpin package[53], which is operated in MATLAB. Typically, ~5 mg of crystal sample was transferred into an EPR tube in a dry-box. After the sample was grinded into crystalline powder by a quartz pestle, the tube was sealed for measurements. A solution with a concentration of $2.98 \times 10^{-3}$ mmol/mL is used for measurements.

**Magnetometry.** Variable-temperature magnetic susceptibility measurements were performed on Quantum Design SQUID-MPMS3 (1–1000 Hz) magnetometer. The experimental susceptibilities were corrected for the diamagnetism of the constituent atoms (Pascal's tables) and background of the sample holder.

**The X-ray absorption fine structure spectroscopy**. Experimental data were recorded at beamline 20 BM of Advanced Photon Source at Argonne National Laboratory, using the Si (111) double crystal monochromator to scan the energy. The spectra were collected in transmission mode and energy calibration were done using Co foil as references. Data analysis was performed at 1W1B and 1W2B, Beijing Synchrotron Radiation Facility.

**Electronic structure**. The G09 program package[54] was used to optimize the structures and for frequency analysis. The model complex for [Co(TPP) (2-MeIm$^-$)]$^-$ and [Co(TPP)(2-MeHIm)] were fully optimized without any constraints by using the Hybrid-GGA functional U-M06[55] and U-B3LYP[56,57].

## Data availability

The X-ray crystallographic coordinates for structures reported in this study have been deposited at the Cambridge Crystallographic Data Center (CCDC), under deposition numbers 1878657-1878660. These data can be obtained free of charge via www.ccdc.cam.ac.uk/data_request/cif. (or from the Cambridge Crystallographic Data Center, 12 Union Road, Cambridge CB21EZ, UK; fax: (+44)1223-336-033; or deposit@ccdc.cam.ac.uk). All the other data that support the findings of this study are available within Supplementary Information and are available from J.L. (jfli@ucas.ac.cn) upon request.

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

## Acknowledgements

We thank the National Natural Science Foundation of China (Grant no. 21771176, U1532128, 21671024, 21702109, 21890722, 11811530637); international cooperation fund and Hundred Talent Program of CAS to J.L.; the Natural Science Foundation of Tianjin City (18JCYBJC21400) and the Fundamental Research Funds for the Central Universities (Nankai University: Grant no. 63191515, 63191523, 63191321) to Q.P.; W.X. is grateful for the fruitful discussion with Y. Joly for using his FDMNES code. The FMS calculations described in this paper (partially) are obtained on the "Era" petascale supercomputer of Computer Network Information Center of Chinese Academy of Sciences. W.X. acknowledges the financial support and the hospitality of LNF under the framework of IHEP&INFN collaboration agreement in 2015–2017. A portion of this work was performed on the Steady High Magnetic Field Facilities, High Magnetic Field Laboratory, CAS. This research used resources of the Advanced Photon Source, an Office of Science User Facility operated for the US DOE Office of Science by Argonne National Laboratory, and was supported by the US DOE under contract number DE-AC02-06CH11357 and by the Canadian Light Source and its funding partners. This work is supported, in part, by the Strategic Priority Research Program of Chinese Academy of Sciences (Grant XDB28000000). We thank BSRF for granting beamtime.

## Author contributions

J.L. conceived and designed the experiments. Jianping Z. and Q.W. performed the experiments. Q.P., H.X., and Z.W. performed the DFT calculations. H.-L.S and F.M. performed the SQUID measurements. W.X., Jiyong Z., and C.-J.S. performed the XAS experiments. Jianping Z., Q.P., Q.W., Z.W., W.X., H.X., H.-L.S., F.M., Jiyong Z., C.-J.S., Jianzhang Z., and J.L. analyzed the data. Jianping Z., Q.P., and J.L. wrote the manuscript. Jianping Z. and Q.P. contributed equally.

## Additional information

**Competing interests:** The authors declare no competing interests.

