## [Transparent Peer Review File · Nature Communications]

Reviewers' comments:

Reviewer #1 (Remarks to the Author):

The manuscript by Jianzhang Zhao, Jianfeng Li and co-workers reports synthesis and physical characterization of Co(II) porphyrins with imidazole and imidazolate axial ligands. These studies reveal a drastic and unexpected difference between in their electronic structure, namely the imidazolate complexes are high-spin. This is confirmed by the crystal structure, EPR, XANES and quantum chemical calculations of total energy, and rationalized by molecular orbitals and geometric arguments. This work seems to be the first case of proton-mediated spin state transition for synthetic porphyrins where the two species (protonated and deprotonated) are isolated and so deeply characterized by different techniques. The results are discussed in the context of analogous spin transitions for Fe(II) and Fe(III) porphyrins and heme enzymes.

The results of this work are novel and clearly relevant for the field of inorganic, bioinorganic, coordination chemistry, and our understanding of heme enzyme mechanisms. I am sure this paper will be read with interest by many researchers in these fields.

The findings of this work may somewhat influence thinking in these fields, mainly in the following aspect: the work clearly shows that a stronger ligand (here, imidazolate vs imidazole) does not always stabilize a lower-spin state. In fact, the contrary is shown in this paper because Co(II) center is "pulled out" of the porphyrin core by strong axial bonding with the imidazolate. The occurrence of this effect is clearly and very nicely demonstrated by this work.

However, I think that despite some similarities, analogies between these Co porphyrins and biologically relevant Fe heme systems are rather limited. A big difference between the two metals is the electronic configuration: d^7 for Co(II) vs d^6 for Fe(II). With one more electron, Co(II) will have d_{z^2} (axial) orbital singly-occupied in both (HS, LS) spin states and hence their relative energies are not really influenced by strength of the axial ligand field. This is very different not the case of Fe(II) or Fe(III) where the d_{z^2} is empty in the LS state and increasing axial field would stabilize the LS state beyond certain point (e.g., 6c Fe(II) complexes with two imidazoles are LS). For Co(II), no matter how strong would be the axial ligand field, the d_{z^2} cannot be depopulated. Thus, the paradox that strong imidazolate ligand stabilize the HS over LS state, which is nicely demonstrated and explained by this study, should be rather considered a "peculiarity" of Co(II) systems (due to the higher number of d electrons and thus "permanent" occupation of the d_{z^2} orbital) and will not be probably observed for Fe(II) porphyrins. Another difference between Co(II) and Fe(II) systems is in the number of exchange interactions stabilizing the HS state with respect to the LS state. This aspect is not explicitly mentioned in the paper, perhaps due to the length limitations.

The results and claims of this paper are generally convincing, although I think that magnetic susceptibility measurements are missing. This is a very simple, yet conclusive technique and it would strengthen the arguments for the change of spin state on deprotonation. Possible trace impurities of [Co(TPP)] mentioned in the EPR study should not affect magnetic susceptibility as much as they do with EPR. Moreover, I think that reporting magnetic moments should be considered "a must" for this type of study, where thorough physicochemical characterization is attempted, and especially when the results are to be published in the prestigious journal.

Moreover, some aspects of the discussion are not clear to me:

>>> p. 9: "Close examinations also found the antibonding interaction between the imidazolate orbital and the β orbital of dyz in LS state. This is compared to the α orbital of dx^2-y^2 in HS state which is 0.364 eV lower in energy."

What does "This" stand for in the second sentence? What is the connection between the two observations being "compared" (antibonding interaction with the β orbital of dyz in LS state and α orbital of dx^2-y^2 in HS state)?

>>> p. 10: "Therefore, it is the strong field of 2-methylimidazolate which has induced the dramatic displacements of Co(II) with the assistance of steric 2-methyl group which lowers the dx^2-y^2 and makes $[(dxy)^2(d_{z^2})^3(dx^2-y^2)^1]$ ($4E_g$) configuration to be accessed."

The role of imidazole deprotonation is clear and well explained by the experiment and calculations. However, it is not clear how the authors evaluated the possible role (or rather "assistance") of steric effect due to the 2-methyl group. If the steric repulsion due to the 2-methyl group is indeed of any importance for the effect of HS stabilization, the authors should justify this conjecture by additional calculations of synthesis of additional complexes without the 2-methyl group. At the moment, the statement about assistance of steric 2-methyl group appears to be pure speculation.

>>> p. 11: "Data availability. The data that support the findings of this study are available from the corresponding author on reasonable request."

What are these additional data and why they are not included in the Supporting Information?

>>> p. 2: "This is the first examples of metalloporphyrins that can switch the spin state through one proton of the proximal ligand."

In fact, there are many heme systems showing experimentally characterized spin transitions on deprotonation, such as microperoxidases. I guess the author meant the first examples among synthetic porphyrin complexes, which could be subject to full physicochemical characterisation (crystal structure, magnetic properties; as is usually not possible for biological systems. However, the above pointed statement can be somewhat misleading.

Also, I believe that it would be right to mention pH-dependent spin state transitions in microperoxidases as important example of spin changes on deprotonation. (See Marques "Insights into porphyrin chemistry provided by the microperoxidases, the haempeptides derived from cytochrome c" Dalton Trans., 2007, 4371-4385, and refs therein)

>>> p. 3 "The unexpected high spin state of imidazolate cobalt(II) porphyrinates, as well as the incomplete spin state transition of iron(II) analogues, arise an interesting question: is the deprotonated imidazole (imidazolate) really stronger than the imidazole?"

This is indeed a very good question and, of course, the authors attempt to answer it in this work. However, I think the correct and most straightforward answer is still missing in the paper: The imidazolate is of course a stronger ligand than imidazole, but for Co(II) porphyrins - due to their d^7 configuration - the dz^2 orbital cannot be depopulated by the increase of the axial ligand field. Instead, the Co(II) ion is significantly displaced out of the porphyrin ring by the imidazolate, leading to the decrease of the equatorial ligand field and stabilization of the HS state. This is why the stronger axial ligand (imidazolate) leads to stabilization of the HS state compared with weaker axial ligand (imidazole).

>>> p. 11 "Given the complex protein environments of heme metal center, the current work implied negative character of proximal ligand does not readily make a complete spin state transition, while big enough metal displacements (e.g. Δ_{24} & $\Delta_4 > 0.2 \text{ \AA}$) always involve with the d orbital reconstruction to perform the physiological functions."

The above sentence is somewhat difficult to understand. What is the evidence that "enough metal displacements (e.g. Δ_{24} & $\Delta_4 > 0.2 \text{ \AA}$) always involve with the d orbital reconstruction"? Is this statement about Fe(II) hemes too? How do the authors relate these statements to "complex protein environments"?

There are also some typos in the manuscript (e.g., page 2 bottom, missing "is"; p. 9 "exanimations"). Also Fig.2 contains unexplained asterisk on the fourth spectrum and there are too many full stops in caption of Fig.1. Finally, ref. 10 has wrong authors' names.

Reviewer #2 (Remarks to the Author):

Comments: "Proton mediated spin state transition of hemes: implications of cobalt analogues"

This article presents the spin state transition (Low Spin to High Spin) of two synthetic cobalt(II) heme systems triggered by the deprotonation of their imidazole axial ligand; and has reported the first instances of proton mediated spin state crossover of metalloporphyrins. Complexes were characterized via Crystallography, EPR and XAS spectroscopy, and complementary computational analysis. Efforts were directed towards understanding the metal-ligand interactions and indicate the importance of metal center displacements (or conformational changes) in spin state transition processes. This article should be published somewhere else due to the lack of impact (or novelty) for a Nature communication, while the conclusions are not justified experimentally as well as they should be, and needs major revisions as outlined in the comments below.

Abstract:

1. Line 3, pag. 1: The authors should give one or two examples of enzymes which do proton mediated spin state transitions
2. Line 3, pag. 2: Authors should state the methods used to determine the spin state of the imidazolate analogues since that is one of the key findings of this paper
3. Line 5, pag. 2: Spelling error: conducted not conducted

Introduction (heading missing in paper): Needs to be revised for grammatical issues

1. Line 13, pag.2: Authors should include references of examples of pH and hydrogen bond mediated spin crossover processes found in the literature
2. Line 17, pag. 2: It is the hydrogen bonding of an nearby Asp residue in peroxidases which imparts anionic character to the proximal histidine --- sentence should be clarified as it reads " the hydrogen bonding of proximal histidine....contributes negative imidazolate character"

Results:

Single Crystal Structures.

1. Line 15, pag. 3: Need to define (TpivPP)
2. Line 20, pag. 3: Reference missing. Where was it reported that the iron(II) analogues need an hindered imidazole to prepare five-coordinate species?
3. Line 21, pag.3: Should define K(222) and TPP and TTP.

It would be nice if the authors could briefly discuss how these novel complexes were prepared and the need for K(222)---presumably needed for solubility?.

This is the first time TPP and TTP are mentioned in the actual text, and they should be defined.

4. Line 8, pag.4: Authors should consider labeling the angle between the imidazole and the closest Nax-Fe-Np plane in Figure 1, as it would enhance the understanding of what the authors are referring to exactly.—and possibly consider elaborating on why a smaller angle corresponds to a more distorted porphyrin core in [Co(TPP)(2-MeHIm)].
5. Line 11, pag. 4: The authors should possibly consider mentioning that the spin state of these two cobalt (II) complexes with neutral imidazole ligands was also verified later on (i.e., further below) with EPR. Again, possibly labels in Figure 1 would aid in the readers understanding of this structural discussion.
6. Line 1, pag. 5: It may be interesting to look at the stereoelectronic effects of the imidazolate axial ligand with various electron withdrawing groups on it or different electron donating groups on

it (possibly also the imidazolate complexes with substituents of different sizes should be characterized to ensure that the complexes are always high-spin regardless of the sterics on the axial ligand).

7. Line 5, pag. 5: References corresponding to those iron(II) analogues in different studies are needed at the end of that sentence.

EPR. ---Figure 2 has a spelling error on x-axis: Gauss not Guass

8. Line 6, pag. 6: Should mention the SI for additional details about the simulations.

9. Line 7, pag. 6: It would be nice if the readers would label the g values on the signals in Figure 2 ---the Figure is in Gauss but the text talks about the peaks in terms of g values

10. Line 10, pag. 6: Should mention the multi-temperature EPR spectra of the cobalt(II) imidazolate complex in the SI (Fig. S5)

11. Line 5, pag. 7: It may be worth adding an extreme excess as 50-100 equivalents of 2-methylimidazolate to see if the reaction goes to completion. It is interesting (or rather surprising) that the imidazolate is a stronger binding ligand (vs. the neutral imidazole ligand) however the Co(II) imidazolate complex never fully forms.

In general, for this EPR section it may also be worth (either in the main text or the SI) putting in the EPR spectra of the cobalt (II) imidazolate TTP derivative, as this is a newly reported species and it would be nice to include the full spectroscopic characterization of this species in this report. Also, it would be nice to see the spin conversion via EPR by this TTP system, since this is a key finding (may be interesting to the reader to see if they completely overlap or if there are any slight differences).

XAS.

Just in terms of formatting in the entire paper and being consistent throughout for every figure, I think it would be better for Figure 3 if the neutral imidazole XANES and EXAFS were on the left and the imidazolate XANES and EXAFS were on the right (to match the crystal structural and EPR figures).

12. Line 11, pag. 8: The authors should explain why it is expected that the HS state has more unoccupied t_{2g} but less unoccupied e_g.

13. Line 12-13, pag. 8: The authors should mention Figures 1 and 2 when tying the XANES back to the previous EPR and crystal structure findings.

It is also of note that the single crystal section never really suggested those findings were consistent of a high spin state for the cobalt(II) imidazolate complex, only that the imidazolate was a stronger binding ligand (vs. the neutral imidazole). It may be worth elaborating back in the single crystal section how those findings suggest a high spin state.

Discussion.

Once again, to be consistent the authors might consider having the 2-MeHIm analysis on the left and the imidazolate analysis on the right for Figure 4.

14. Line 20, pag. 8: "... as seen in the alkaline transition." This sentence needs references.

15. Line 5, pag. 9: Spelling error --- examinations not "exanimations"

16. Line 5 and Line 13, pag. 9: should mention Figure 1 or Table 1 to tie this discussion back to the aforementioned results

17. Line 5, pag. 10: I think the authors should make the [Co(TPP)(Im-)]- complex and look at the structural features via a crystal structure or do DFT analysis to substantiate the contribution of the 2-methyl group to facilitate spin crossover. It would be interesting to see if the spin crossover will only occur if the 2-methyl group is present.

18. Line 4, pag.11: The conclusion states that “comprehensive spectroscopic characterization was done on four isolated cobalt(II) heme complexes.....”, however in the main text and in the SI the EPR, XAS, or DFT data for the TTP imidazole and imidazolate complexes are not present (only the crystal structure data). I do not think the authors can make this conclusion without providing the rest of this data--- only comprehensive spectroscopic characterization was provided for 2 cobalt(II) heme complexes. I also do not think they can say the TTP complexes also do proton mediated spin state changes based only on the crystal structure data without the key EPR and XAS data that really substantiate their hypothesis.

19. The last paragraph on pg. 11 has grammatical issues.

Reviewer #3 (Remarks to the Author):

While the scientific content of this paper is interesting, I find difficulties with the presentation of the work and, importantly, I believe that there are some problems with the EPR data presented and its interpretation.

Firstly, the data in figure 2 is measured at multiple different temperatures, as it has previously been shown that temperature can initiate the low to high spin transition in related compounds ('The characterization of cobalt(II) derivatives of selected substituted meso-tetraphenyl and tetrapyrrolyl porphyrins by EPR spectroscopic study' by D. Skrzypek, I. Madejska and J. Havdas Solid State Sciences Volume 9, Issues 3–4, March–April 2007, Pages 295-302) I think it is vital that the ESR data presented to illustrate the high to low spin transition due to the effect of deprotonation be presented at a single temperature and variable temperature EPR to be shown for all compounds (similar to that presented for $[K(222)][Co(TPP)(2-MeIm-)]$ in figure S5) to verify that the high to low spin transition is not an artefact of temperature.

Secondly the ESR data in figure S6 is badly saturated making it un-interpretable, I think that it should be re-measured before publication can be considered. While I agree with the overall conclusions drawn in terms of the change from low to high spin on deprotonation, I do not agree with the interpretation of all of the results from the ESR data, as detailed below in my comments on the EPR results section.

Addressing the issues of presentation; the style of this paper does not match Nature Communications style - it is entirely missing a methods section in the main text. The methods are only given in the SI, and even then they are incomplete for example no information is provided about the concentrations used for the EPR measurements in frozen solution or the grinding process used to generate the crystalline samples.

Furthermore, there is no distinction between the abstract and what I assume is the introduction; it seems that there is an introduction heading missing from between the 1st and second paragraphs

on page 2? It is my belief that in order for this paper to be considered for publication in Nature communications, it should be significantly rewritten to match the style of this journal.

As a native English speaker I found this paper very difficult to read. There are several places where the sentence structure is confusing, tenses and articles are missing or wrong and the author's usage of 'thus' is incorrect. I have attempted to detail these, along with my concerns over the ESR data presented in my review.

Abstract:

The form of this abstract bears a lot of similarity to reference 10 (Sahoo, D. et al. Hydrogen-Bonding Interactions Trigger a Spin-Flip in Iron(III) Porphyrin Complexes. *Angew. Chem., Int. Ed.* 54, 4796-4800 (2015).) While the work in reference 10 is on an Iron(III) complex and therefore bears some similarity to the native Heme in Cytochrome P450, Cytochrome P450 does not contain cobalt and only a relatively small number of studies have incorporated cobalt into Cytochrome P450 (such as: *Methods Mol Biol.* 2013;987:107-13. doi: 10.1007/978-1-62703-321-3_9. Expression in *Escherichia coli* of a cytochrome P450 enzyme with a cobalt protoporphyrin IX prosthetic group. Straub WE, Nishida CR, de Montellano PR.). For this reason the use of 'thus' at the start of the third sentence is in my opinion is incorrect. Either the specific link to cytochrome P450 should be removed or an additional sentence of explanation as to why the work in this paper is specifically relevant to P450 is needed as cobalt containing enzymes such as carbonic anhydrase also undergo spin transitions.

Introduction(?)

- The notation '(5c, HS towards 6c, LS)' is unclear and not explained, all abbreviations should be given in full the first time they are used.
- No reference is given for the statement: 'Besides this, spin state transition can also be controlled by (weak)external perturbations such as pH and hydrogen bonding.'
- 'The proximal His18 of Cytochrome c...' it is not clear which form of cytochrome c is being referred to here.
- '(HS towards pH > 11, LS)' notation not clear.
- 'Although the alkaline transition have been studied...' should be Although the alkaline transition has been studied...
- 'This is comparable to the hydrogen bonding of proximal histidine in oxygen activating heme enzymes...' I am not clear what is comparable in this sentence, the authors need to be more specific.
- '... perturbations still poorly understood.' Should be '... perturbations is still poorly understood.'
- 'Porphyrin system has been used to investigate the impact of external perturbations for various advantages including the isolable active site where the spin state transition happened.' Does not make sense as a sentence - the tenses are all over the place and no references are given.
- '...imidazolate has been developed by complete deprotonation of the imidazole and accepted as a stronger field ligand for its better σ and n donation.' Accepted in what sense? By the scientific community or by the metal centre? This sentence needs rewording for clarity.
- 'Mossbauer characterizations revealed two different HS configurations...' HS is used without any definition and needs to be defined as an abbreviation.
- '...arise an interesting question:' should be '...pose an interesting question:'.

Results (Single crystal structures.)

- '...the starting material are imidazole cobalt(II) porphyrinates, the species of which are all five-coordinate due to the destabilization of the six-coordinate compound by singly populated dz² orbital.' It is not clear which species you refer to here please be specific.
- Missing article: '...compound by singly populated dz² orbital.' should be '...compound by the singly populated dz² orbital.'
- 'This is contrasted to the iron(II) analogues' needs a reference.
- 'This is contrasted to the iron(II) analogues, the use of hindered imidazole is necessary to prepare five-coordinate species, all of which known so far are high-spin state (3d⁶, S = 2).' The two halves

of this sentence bear no relationship to one another and the ideas should be divided into two sentences for clarity, and references given.

·[K(222)]requires a definition.

·I am unclear why 'Nax-Fe-Np plane' is referred to for a Cobalt compound - I presume it should read 'Nax-Co-Np plane'?

·' $\Delta 24$ & $\Delta 4 < 0.26$ Angstrom).' Should read ' $\Delta 24$ & $\Delta 4 \leq 0.26$ Angstrom).' According to table 1

·'(> 0.43 Angstrom).' Should read ' ≥ 0.43 Angstrom).' According to table 1

·'Moreover, the two imidazolate ligand show shorter...' should read 'Moreover, the two imidazolate ligands show shorter...'

·'...than the imidazole (> 2.1 Angstrom), indicating its stronger bonding.' I am unclear as to what 'its' refers to in this sentence please be specific.

Table 1:

·'[Fe(TPP)(2-MeHIm)]', should read '[Fe(TPP)(2-MeHIm)].1.5C6H5Cl' for accuracy as other structures exist under other conditions.

·Two entries in the table are missing references and other information - if information is not available missing information should, for clarity, be indicated explicitly rather than blank spaces left.

Figure 2:

·The labelling of this figure is unsatisfactory.

·As the paper 'The characterization of cobalt(II) derivatives of selected substituted meso-tetraphenyl and tetrapyrrolyl porphyrins by EPR spectroscopic study' by D. Skrzypek, I. Madejska and J. Havdas Solid State Sciences Volume 9, Issues 3–4, March–April 2007, Pages 295-302 shows that the low spin to high spin transition can be achieved in similar porphyrin compounds – I think that the data presented in Figure 2 needs to be obtained at the same temperature and variable temperature ESR for all compounds shown in the SI to verify that the effects seen are indeed due to the ligands and not due to acquisition of the data at different temperatures.

·I think that it would be a lot clearer to write the identities of the chemical compounds studied on the figure rather than only in the legend.

·Simulations are only provided for 2 of the 4 spectra and no simulation parameters are given - without these the simulations are meaningless. The simulation parameters need to be listed in full even if it is in the SI.

·The * in the '1.8 K THF solution' spectra is not referred to in the figure legend, or explained - an explanation for this additional signal is needed

·reference 36 is not a reference it is a foot note - if these are not allowed in the style of this journal this information must be included in the main text or figure legend and the effect of this noted in the spectra if it is observed. Also I think 'The grind process to make the crystalline samples...' should be 'The grinding process to make the crystalline samples...'

·The simulation presented for the imidazolate system is not a very good match to the experimental data - is there a reason for this? Could a 2nd species be present? Or can the simulation be improved?

Results (Electron Paramagnetic Resonance (EPR).)

·The complete parameter set used in the EPR simulations must be given somewhere in the paper or SI.

·No information is provided about the sample conditions used for the EPR acquisition, concentrations for the solution state samples, or the grinding process used to make the crystalline samples. This needs to be included in the paper.

·'...shows axial signals at $g_{\text{perp}} = 2.3$, $g_{\text{||}} = 2.0$ ($A_{\text{Co}} = 79.2$ G), which are typical...' would better read '...shows axial symmetry, with $g_{\text{perp}} = 2.3$, $g_{\text{||}} = 2.0$ ($A_{\text{Co}} = 79.2$ G), which is typical...'

·You quote the characteristic resonances for the imidazolate compound as 6.0, 4.0 and 2.0, are these g values? If so they should be characterized as g_x , g_y and

g_z . You then go onto give ' $A_{Co} = 82.0 \text{ G}$ ' as it is usually expected that the g and A frames are coincident unless otherwise expressed, as such this notation is not correct here as g_x is not equal to g_y . The hyperfine values must be listed either in the same reference frame as the g tensor or if the reference frames are not the same the angles between them should be specified - these values should be easily extractable from your easyspin simulations.

'...excited state and resonances at 4.0 and 2.0 come...' and '...weak signal is observed at ~ 2.3 ...' again are these g values? Please specify or give the plots in figure 2 on a g axis scale.

You state 'In both phases a weak signal is observed at ~ 2.3 which might correspond to the four-coordinate [Co(TPP)]' I do not agree with this. I would like to draw your attention to the paper: Probing the Surrounding of a Cobalt(II) Porphyrin and its Superoxo Complex by EPR Techniques by M. Baumgarten, C. J. Winscom, and W. Lubitz (Applied Magnetic Resonance February 2001, Volume 20, Issue 1-2, pp 35-70). In this work they study [Co(OEP)] in THF using ESR. They found that in this solvent the [Co(OEP)] could either remain 4 coordinate (base-off) or interact with the solvent to form a base-on state, both of which have very different ESR spectra (see figure 3 of the M. Baumgarten, C. J. Winscom, and W. Lubitz paper). In addition neither of these species were found to have a g value of 2.3 (See table 1 of the M. Baumgarten, C. J. Winscom, and W. Lubitz paper). The pure four coordinate spectrum has also been observed for Co(TPP) in a crystalline solid, and does not have $g=2.3$; as seen in Solid State Sciences Volume 9, Issues 3-4, March-April 2007, Pages 295-302 The characterization of cobalt(II) derivatives of selected substituted meso-tetraphenyl and tetrapyrrolyl porphyrins by EPR spectroscopic study by D. Skrzypek, I. Madejska and J. Havdas. I therefore do not believe that the signal you are observing at $g=2.3$ in the imidazolate samples is 4 coordinate [Co(TPP)]. In my mind it is more likely incompletely deprotonated sample and thus a residue signal from the imidazole compound. The reference spectra with 0 equivalents of imidazolate in THF is so badly saturated and therefore distorted (see my comment below about figure S6) that it is impossible in my opinion to use this as a comparison to the other datasets with larger equivalents of imidazolate.

'Hence, the high-spin resonances is assigned...' would read better 'Hence, the high-spin resonances are assigned...'

Figure S6:

This figure is not suitable for publication - the spectra at 0 equivalents is clearly saturated - all of the signal is positive, whereas in continuous wave ESR the signal should be collected as a dispersive signal. It is likely that the problem could be relieved by increasing the temperature and reducing the microwave power used in the measurements.

Unfortunately presentation of such data also makes me question the shape of the signals seen in the 3 and 7 equivalent spectra where the flat-tops of the low field feature look partially reminiscent truncated signals, I believe that this needs to be checked by the authors.

All of the data in this figure needs to be re-measured at a higher temperature and a microwave power dependence study carried out for each set of conditions to ensure that the signals presented are not saturated.

Results (X-ray Absorption Spectroscopy):

X-ray Absorption Spectroscopy is outside my field of expertise and therefore I do not feel able to comment on this section.

Discussion:

I do not understand the final sentence, 'Given the complex protein environments of heme metal center, the current work implied negative character of proximal ligand does not readily make a complete spin state transition, while big enough metal displacements (e.g. $\Delta 24$ & $\Delta 4 > 0.2$ Angstrom) always involve with the d orbital reconstruction to perform the physiological functions.' Please reword.

Data availability:

Nature journals mandate the deposition of small molecule crystal structures in the Cambridge

Structural Database, this should be done and the data availability statement corrected accordingly.

References:

- The format of reference 10 is not correct.
- 36 is a footnote not a reference - please remove and put the information into the main text.
- 41 is a footnote not a reference - please remove and put the information into the main text.

Reviewer 1

1. although I think that magnetic susceptibility measurements are missing. This is a very simple, yet conclusive technique and it would strengthen the arguments for the change of spin state on deprotonation. Possible trace impurities of [Co(TPP)] mentioned in the EPR study should not affect magnetic susceptibility as much as they do with EPR. Moreover, I think that reporting magnetic moments should be considered “a must” for this type of study, where thorough physicochemical characterization is attempted, and especially when the results are to be published in the prestigious journal.

Thanks for the comments.

We have measured the magnetic susceptibility. The results of HS and LS states for imidazolate and imidazole species respectively are given in the text.

2. >>> p. 9: “Close examinations also found the antibonding interaction between the imidazolate orbital and the β orbital of d_{yz} in LS state. This is compared to the α orbital of $d_{x^2-y^2}$ in HS state which is 0.364 eV lower in energy.”
What does “This” stand for in the second sentence? What is the connection between the two observations being “compared” (antibonding interaction with the β orbital of d_{yz} in LS state and α orbital of $d_{x^2-y^2}$ in HS state)?

Thanks for the comments. This has been modified to:

Close examinations also found the anti-bonding interaction between the imidazolate π orbital and the β orbital of d_{yz} in LS state, which is 0.364 eV higher in energy than the α orbital of $d_{x^2-y^2}$ in HS state (grey arrow).

3. >>> p. 10: “Therefore, it is the strong field of 2-methylimidazolate which has induced the dramatic displacements of Co(II) with the assistance of steric 2-methyl group which lowers the $d_{x^2-y^2}$ and makes [(d_{xy})²(d_{z^2})¹($d_{x^2-y^2}$)¹] (4Eg) configuration to be accessed.” The role of imidazole deprotonation is clear and well explained by the experiment and calculations. However, it is not clear how the authors evaluated the possible role (or rather “assistance”) of steric effect due to the 2-methyl group. If the steric repulsion due to the 2-methyl group is indeed of any importance for the effect of HS stabilization, the authors should justify this conjecture by additional calculations of synthesis of additional complexes without the 2-methyl group. At the moment, the statement about assistance of steric 2-methyl group appears to be pure speculation.

Thanks for your comments.

Non-hindered *imidazolate* (i.e. 4-MeIm⁻) ligated Co(II) porphyrinates have been isolated and studied in this Lab. These complexes are all six-coordinate LS with near-zero metal displacements (Figures), contrasted to the five-coordinate HS 2-MeIm⁻ derivatives. The work are underway and will be reported in a separate paper.

Hence, the HS 2-MeIm⁻ derivatives are stabilized for the combination of both the steric

Thermal ellipsoid diagrams of $[K(222)]_2[Co(TPP)(4-MeIm^-)_2]$ and $[K(222)]_2[Co(TMP)(4-MeIm^-)_2]$. Hydrogen atoms, $[K(222)]^+$ and solvent molecules are not shown for clarity.

2-Methyl group which makes the five-coordinate available and the strong *imidazolate* bonding which pulls the metal out of plane significantly. We have discussed this in the 2ed paragraph of the “discussion” part.

4. p. 11: “Data availability. The data that support the findings of this study are available from the corresponding author on reasonable request.”

Thanks for your comments. This has been corrected.

5. >>> p. 2: “This is the first examples of metalloporphyrins that can switch the spin state through one proton of the proximal ligand.” In fact, there are many heme systems showing experimentally characterized spin transitions on deprotonation, such as microperoxidases. I guess the author meant the first examples among synthetic porphyrin complexes, which could be subject to full physicochemical characterisation (crystal structure, magnetic properties; as is usually not possible for biological systems. However, the above pointed statement can be somewhat misleading.

Also, I believe that it would be right to mention pH-dependent spin state transitions in microperoxidases as important example of spin changes on deprotonation. (See Marques “Insights into porphyrin chemistry provided by the microperoxidases, the haempeptides derived from cytochrome c” Dalton Trans., 2007, 4371-4385, and refs therein)

Thanks for your comments.

“This is the first examples of metalloporphyrins that can switch the spin state through one proton of the proximal ligand.” has been changed to

“This is the first examples of synthetic metalloporphyrins that can switch the spin state through one proton of the proximal ligand.”

“pH-dependent spin state transitions in microperoxidases” has been added in the Introduction, and the mentioned paper has been cited in the text.

6. p. 3 “The unexpected high spin state of imidazolate cobalt(II) porphyrinates, as well as the

incomplete spin state transition of iron(II) analogues, arise an interesting question: is the deprotonated imidazole (imidazolate) really stronger than the imidazole?”

This is indeed a very good question and, of course, the authors attempt to answer it in this work. However, I think the correct and most straightforward answer is still missing in the paper: The imidazolate is of course a stronger ligand than imidazole, but for Co(II) porphyrins - due to their d7 configuration - the dz2 orbital cannot be depopulated by the increase of the axial ligand field. Instead, the Co(II) ion is significantly displaced out of the porphyrin ring by the imidazolate, leading to the decrease of the equatorial ligand field and stabilization of the HS state. This is why the stronger axial ligand (imidazolate) leads to stabilization of the HS state compared with weaker axial ligand (imidazole).

Thanks for your comments. We have discussed this in the 2ed paragraph of the “discussion” part.

7. >>> p. 11 “Given the complex protein environments of heme metal center, the current work implied negative character of proximal ligand does not readily make a complete spin state transition, while big enough metal displacements (e.g. Δ_{24} & $\Delta_4 > 0.2 \text{ \AA}$) always involve with the d orbital reconstruction to perform the physiological functions.” The above sentence is somewhat difficult to understand. What is the evidence that “enough metal displacements (e.g. Δ_{24} & $\Delta_4 > 0.2 \text{ \AA}$) always involve with the d orbital reconstruction”? Is this statement about Fe(II) hemes too? How do the authors relate these statements to “complex protein environments”?

Thanks for your comments. The sentence has been modified.

8. There are also some typos in the manuscript (e.g., page 2 bottom, missing “is”; p. 9 “exanimations”). Also Fig.2 contains unexplained asterisk on the fourth spectrum and there are too many full stops in caption of Fig.1. Finally, ref. 10 has wrong authors’ names.

Thanks for your comments. These have been corrected.

The explanation on the asterisk of Fig.2 is given in the text.

Reviewer 2

1. 1. Line 3, pag. 1: The authors should give one or two examples of enzymes which do proton mediated spin state transitions
Line 3, pag. 2: Authors should state the methods used to determine the spin state of the imidazolate analogues since that is one of the key findings of this paper.
Line 5, pag. 2: Spelling error: conducted not conducted

Thanks for your comments. These have been corrected and modified in the text.

2. Line 13, pag.2: Authors should include references of examples of pH and hydrogen bond mediated spin crossover processes found in the literature
Line 17, pag. 2: It is the hydrogen bonding of an nearby Asp residue in peroxidases which imparts anionic character to the proximal histidine --- sentence should be clarified as it reads “ the hydrogen bonding of proximal histidine....contributes negative imidazolate character”

Thanks for your comments. The references have been given.

The sentence has been modified “This is comparable to the hydrogen bonding between Asp and proximal histidine in oxygen activating heme enzymes (i.e. peroxidases) which contributes to negative imidazolate character to promote heterolytic cleavage of the O–O bond to form a ferryl species”.

3. Line 15, pag. 3: Need to define (TpivPP) 2. Line 20, pag. 3: Reference missing. Where was it reported that the iron(II) analogues need an hindered imidazole to prepare five-coordinate species? Line 21, pag.3: Should define K(222) and TPP and TTP. It would be nice if the authors could briefly discuss how these novel complexes were prepared and the need for K(222)---presumably needed for solubility?. This is the first time TPP and TTP are mentioned in the actual text, and they should be defined.

Thanks for your comments. All the abbreviations have been defined and given in the text. The reference about hindered imidazole to prepare five-coordinate species is given. Synthesis method is discussed. Cryptand 222 (Crypt-222, 4, 7, 13, 16, 21, 24-Hexaoxa-1, 10-diazabicyclo[8.8.8]hexacosane) is used to stabilized the K⁺ cation.

4. 4. Line 8, pag.4: Authors should consider labeling the angle between the imidazole and the closest Nax-Fe-Np plane in Figure 1, as it would enhance the understanding of what the authors are referring to exactly—and possibly consider elaborating on why a smaller angle corresponds to a more distorted porphyrin core in [Co(TPP)(2-MeHIm)].

Thanks for your comments. Since the ORTEP diagrams in Figure 1 are side-on views, it is hard to clearly elaborate the φ angles (the angle between the imidazole and the closest Nax-Co-Np plane). However, the φ angle is illustrated in the 24-atom mean plane of Figure S3. In addition, modification has been made in the text to elaborate on why a smaller angle corresponds to a more distorted porphyrin core.

5. 5. Line 11, pag. 4: The authors should possibly consider mentioning that the spin state of these two cobalt (II) complexes with neutral imidazole ligands was also verified later on (i.e., further below) with EPR. Again, possibly labels in Figure 1 would aid in the readers understanding of this structural discussion.

Thanks for your comments. The spin state change has been labeled in Figure 1.

6. 6. Line 1, pag. 5: It may be interesting to look at the stereoelectronic effects of the

imidazolate axial ligand with various electron withdrawing groups on it or different electron donating groups on it (possibly also the imidazolate complexes with substituents

Thermal ellipsoid diagrams of $[K(222)]_2[Co(TPP)(4-MeIm)_2]$ and $[K(222)]_2[Co(TMP)(4-MeIm)_2]$. Hydrogen atoms, $[K(222)]^+$ and solvent molecules are not shown for clarity.

of different sizes should be characterized to ensure that the complexes are always high-spin regardless of the sterics on the axial ligand).

Thanks for your comments. Non-hindered *imidazolate* (i.e. $4-MeIm^-$) ligated Co(II) porphyrinates have been isolated and studied in this Lab. These complexes are all six-coordinate LS with near-zero metal displacements (Figures), contrasted to the five-coordinate HS $2-MeIm^-$ derivatives. The work are underway and will be reported in a separate paper.

7. Line 5, pag. 5: References corresponding to those iron(II) analogues in different studies are needed at the end of that sentence.

Thanks for your comments. The references are added.

8. EPR. ---Figure 2 has a spelling error on x-axis: Gauss not Guass. Line 6, pag. 6: Should mention the SI for additional details about the simulations. Line 7, pag. 6: It would be nice if the readers would label the g values on the signals in Figure 2 ----the Figure is in Gauss but the text talks about the peaks in terms of g values. Line 10, pag. 6: Should mention the multi-temperature EPR spectra of the cobalt(II) imidazolate complex in the SI (Fig. S5)

Thanks for your comments. typos are corrected. g values are given in Figures. Additional details about the simulations and multi-temperature EPR spectra are given in text.

9. Line 5, pag. 7: It may be worth adding an extreme excess as 50-100 equivalents of 2-methylimidazolate to see if the reaction goes to completion. It is interesting (or rather surprising) that the imidazolate is a stronger binding ligand (vs. the neutral imidazole ligand) however the Co(II) imidazolate complex never fully forms.

In general, for this EPR section it may also be worth (either in the main text or the SI) putting in the EPR spectra of the cobalt (II) imidazolate TTP derivative, as this is a newly

reported species and it would be nice to include the full spectroscopic characterization of this species in this report. Also, it would be nice to see the spin conversion via EPR by this TTP system, since this is a key finding (may be interesting to the reader to see if they completely overlap or if there are any slight differences).

Thanks for your comments. 7 equivalents of 2-methylimidazolate has saturated the reaction solution. We have performed EPR and magnetic studies on TTP system which shows identical results with TPP counterparts. These are given and discussed in the text and SI.

10. XAS. Just in terms of formatting in the entire paper and being consistent throughout for every figure, I think it would be better for Figure 3 if the neutral imidazole XANES and EXAFS were on the left and the imidazolate XANES and EXAFS were on the right (to match the crystal structural and EPR figures).

Thanks for your comments. All the figures are reorganized with neutral imidazole derivatives on the left sides.

11. Line 11, pag. 8: The authors should explain why it is expected that the HS state has more unoccupied t_{2g} but less unoccupied e_g. Line 12-13, pag. 8: The authors should mention Figures 1 and 2 when tying the XANES back to the previous EPR and crystal structure findings.

Thanks for your comments. These have been modified in the text.

12. It is also of note that the single crystal section never really suggested those findings were consistent of a high spin state for the cobalt(II) imidazolate complex, only that the imidazolate was a stronger binding ligand (vs. the neutral imidazole). It may be worth elaborating back in the single crystal section how those findings suggest a high spin state.

Thanks for your comments. It has been generally accepted in the metalloporphyrin chemistry that the longer (M–N_p)_{av} and larger metal out of plane displacements usually suggest the high spin state (Scheidt, W. R. & Reed, C. A. Spin-state/stereochemical relationships in iron porphyrins: implications for the hemoproteins. *Chem. Rev.* **81**, 543-555 (1981)).

13. Once again, to be consistent the authors might consider having the 2-MeHIm analysis on the left and the imidazolate analysis on the right for Figure 4.

Thanks for your comments. The Figures have been reorganized.

14. Line 20, pag. 8: "... as seen in the alkaline transition." This sentence needs references. 15. Line 5, pag. 9: Spelling error --- examinations not "exanimations". 16. Line 5 and Line 13, pag. 9: should mention Figure 1 or Table 1 to tie this discussion back to the

aforementioned results.

Thanks for your comments. Corrections and modifications have been made.

15. 17. Line 5, pag. 10: I think the authors should make the [Co(TPP)(Im⁻)]- complex and look at the structural features via a crystal structure or do DFT analysis to substantiate the contribution of the 2-methyl group to facilitate spin crossover. It would be interesting to see if the spin crossover will only occur if the 2-methyl group is present.

Thanks for your comments. Non-hindered *imidazolate* (i.e. 4-MeIm⁻) ligated Co(II) porphyrinates have been isolated and studied in this Lab. These complexes are all six-coordinate LS with near-zero metal displacements, contrasted to the five-coordinate HS 2-MeIm⁻ derivatives. The work are underway and will be reported in a separate paper.

16. Line 4, pag.11: The conclusion states that “comprehensive spectroscopic characterization was done on four isolated cobalt(II) heme complexes.....”, however in the main text and in the SI the EPR, XAS, or DFT data for the TTP imidazole and imidazolate complexes are not present (only the crystal structure data). I do not think the authors can make this conclusion without providing the rest of this data--- only comprehensive spectroscopic characterization was provided for 2 cobalt(II) heme complexes. I also do not think they can say the TTP complexes also do proton mediated spin state changes based only on the crystal structure data without the key EPR and XAS data that really substantiate their hypothesis.

Thanks for your comments. The spectroscopic characterization on TTP have been performed and given in the text which does show expected spin state changes.

17. The last paragraph on pg. 11 has grammatical issues.

Thanks for your comments. This paragraph has been modified.

Reviewer 3

1. Firstly, the data in figure 2 is measured at multiple different temperatures, as it has previously been shown that temperature can initiate the low to high spin transition in related compounds (“The characterization of cobalt(II) derivatives of selected substituted meso-tetraphenyl and tetrapyrrolyl porphyrins by EPR spectroscopic study’ by D. Skrzypek, I. Madejska and J. Havdas Solid State Sciences Volume 9, Issues 3–4, March–April 2007, Pages 295-302) I think it is vital that the ESR data presented to illustrate the high to low spin transition due to the effect of deprotonation be presented at a single temperature and variable temperature EPR to be shown for all compounds (similar to that

presented for [K(222)][Co(TPP)(2-MeIm⁻)] in figure S5) to verify that the high to low spin transition is not an artefact of temperature.

Thanks for your comments. Multitemperature EPR and Magnetic measurements have been done and given in the text, which verify that the high to low spin transition is not an artefact of temperature.

2. Secondly the ESR data in figure S6 is badly saturated making it un-interpretable, I think that it should be re-measured before publication can be considered. While I agree with the overall conclusions drawn in terms of the change from low to high spin on deprotonation, I do not agree with the interpretation of all of the results from the ESR data, as detailed below in my comments on the EPR results section.

Thanks for your comments. Figure S6 has been re-measured and given.

3. Addressing the issues of presentation; the style of this paper does not match Nature Communications style - it is entirely missing a methods section in the main text. The methods are only given in the SI, and even then they are incomplete for example no information is provided about the concentrations used for the EPR measurements in frozen solution or the grinding process used to generate the crystalline samples. Furthermore, there is no distinction between the abstract and what I assume is the introduction; it seems that there is an introduction heading missing from between the 1st and second paragraphs on page 2? It is my belief that in order for this paper to be considered for publication in Nature communications, it should be significantly rewritten to match the style of this journal.

Thanks for your comments. A detailed methods section including the concentrations of the EPR measurements in frozen solution and the grinding process has been given in the text. Sorry for the missing of the “introduction heading”. The whole manuscript has been significantly modified to match the style of the journal.

Abstract

4. The form of this abstract bears a lot of similarity to reference 10 (Sahoo, D. et al. Hydrogen-Bonding Interactions Trigger a Spin-Flip in Iron(III) Porphyrin Complexes. *Angew. Chem., Int. Ed.* 54, 4796-4800 (2015).) While the work in reference 10 is on an Iron(III) complex and therefore bears some similarity to the native Heme in Cytochrome P450, Cytochrome P450 does not contain cobalt and only a relatively small number of studies have incorporated cobalt into Cytochrome P450 (such as: *Methods Mol Biol.* 2013;987:107-13. doi: 10.1007/978-1-62703-321-3_9. Expression in Escherichia coli of a cytochrome P450 enzyme with a cobalt protoporphyrin IX prosthetic group. Straub WE, Nishida CR, de Montellano PR.). For this reason the use of 'thus' at the start of the third sentence in my opinion is incorrect. Either the specific link to cytochrome P450 should be removed or an additional sentence of explanation as to why the work in this paper is specifically relevant to P450 is needed as cobalt containing enzymes such as carbonic

anhydrase also undergo spin transitions.

Thanks for your comments. The abstract has been modified and “thus” is removed.

Introduction

5. The notation '(5c, HS towards 6c, LS)' is unclear and not explained, all abbreviations should be given in full the first time they are used. No reference is given for the statement: 'Besides this, spin state transition can also be controlled by (weak)external perturbations such as pH and hydrogen bonding.'

Thanks for your comments. All abbreviations and the missed reference have been given.

6. 'The proximal His18 of Cytochrome c...' it is not clear which form of cytochrome c is being referred to here.

Thanks for your comments. This information has been given.

7. '(HS towards pH > 11, LS)' notation not clear.

Thanks for your comments. The sentence has been modified.

8. 'Although the alkaline transition have been studied...' should be 'Although the alkaline transition has been studied...'

This is comparable to the hydrogen bonding of proximal histidine in oxygen activating heme enzymes...' I am not clear what is comparable in this sentence, the authors need to be more specific.

'... perturbations still poorly understood.' Should be '... perturbations is still poorly understood.'

'Porphyrin system has been used to investigate the impact of external perturbations for various advantages including the isolable active site where the spin state transition happened.' Does not make sense as a sentence - the tenses are all over the place and no references are given.

'...imidazolate has been developed by complete deprotonation of the imidazole and accepted as a stronger field ligand for its better σ and π donation.' Accepted in what sense? By the scientific community or by the metal centre? This sentence needs rewording for clarity.

'Mossbauer characterizations revealed two different HS configurations...' HS is used without any definition and needs to be defined as an abbreviation.

'...arise an interesting question:' should be '...pose an interesting question:'.

Thanks for your comments. Corrections, modifications and references have been given.

Results (Single crystal structures.)

9. '...the starting material are imidazole cobalt(II) porphyrinates, the species of which are all

five-coordinate due to the destabilization of the six-coordinate compound by singly populated dz² orbital.’ It is not clear which species you refer to here please be specific.

·Missing article: ‘...compound by singly populated dz² orbital.’ should be ‘...compound by the singly populated dz² orbital.’

·‘This is contrasted to the iron(II) analogues’ needs a reference.

·‘This is contrasted to the iron(II) analogues, the use of hindered imidazole is necessary to prepare five-coordinate species, all of which known so far are high-spin state (3d⁶, S = 2).’ The two halves of this sentence bear no relationship to one another and the ideas should be divided into two sentences for clarity, and references given.

·[K(222)]requires a definition. ·I am unclear why ‘Nax-Fe-Np plane’ is referred to for a Cobalt compound - I presume it should read ‘Nax-Co-Np plane’?

·‘(Δ₂₄ & Δ₄ < 0.26 Angstrom).’ Should read (Δ₂₄ & Δ₄ ≤ 0.26 Angstrom).’ According to table 1

·‘Moreover, the two imidazolate ligand show shorter...’ should read ‘Moreover, the two imidazolate ligands show shorter...’

·‘(> 0.43 Angstrom).’ Should read ‘(≥ 0.43 Angstrom).’ According to table 1

·‘...than the imidazole (> 2.1 Angstrom), indicating its stronger bonding.’ I am unclear as to what ‘its’ refers to in this sentence please be specific.

Thanks for your comments. Corrections, modifications and references have been given.

Table 1:

10. ·[Fe(TPP)(2-MeHIm)]’, should read ‘[Fe(TPP)(2-MeHIm)].1.5C₆H₅Cl’ for accuracy as other structures exist under other conditions.

·Two entries in the table are missing references and other information - if information is not available missing information should, for clarity, be indicated explicitly rather than blank spaces left.

Thanks for your comments. We surely understood your concerning on the accuracy of complex names. However several reasons make us feel the current type setting are appropriate. 1. Most of the complexes in Table 1 have solvent molecules, it is difficult to have all these solvent names in one table. 2. The references in the last column of the table make the complex name in each entry solely. 3. Usually the solvent molecule is not included in the table when comparing the structural parameters, for example in Nat. Commun. 3, 720 (2012). doi:10.1038/ncomms1718.

“NA” has been indicated in the table to note not available information. The missed references are given.

Figure 2:

11. ·The labelling of this figure is unsatisfactory. ·As the paper ‘The characterization of cobalt(II) derivatives of selected substituted meso-tetraphenyl and tetrapyrrolyl porphyrins by EPR spectroscopic study’ by D. Skrzypek, I. Madejska and J. Havdas Solid State Sciences Volume 9, Issues 3–4, March–April 2007, Pages 295-302 shows that the low spin to high spin transition can be achieved in similar porphyrin compounds – I think that

the data presented in Figure 2 needs to be obtained at the same temperature and variable temperature ESR for all compounds shown in the SI to verify that the effects seen are indeed due to the ligands and not due to acquisition of the data at different temperatures.

Thanks for your comments. EPR experiments at the same temperature (~ 2 K) have been re-measured. Variable temperature EPR for all compounds have been re-measured and given in the SI.

12. ·I think that it would be a lot clearer to write the identities of the chemical compounds studied on the figure rather than only in the legend.
·Simulations are only provided for 2 of the 4 spectra and no simulation parameters are given - without these the simulations are meaningless. The simulation parameters need to be listed in full even if it is in the SI.
·The * in the '1.8 K THF solution' spectra is not referred to in the figure legend, or explained - an explanation for this additional signal is needed
·reference 36 is not a reference it is a foot note - if these are not allowed in the style of this journal this information must be included in the main text or figure legend and the effect of this noted in the spectra if it is observed. Also I think 'The grind process to make the crystalline samples...' should be 'The grinding process to make the crystalline samples...'

Thanks for your comments. Simulation parameters are given in SI.

The explanation on the asterisk of Fig.2 is given in the text.

Corrections and modifications have been given according to your suggestions.

Results (Electron Paramagnetic Resonance (EPR).)

13. ·The complete parameter set used in the EPR simulations must be given somewhere in the paper or SI.
·No information is provided about the sample conditions used for the EPR acquisition, concentrations for the solution state samples, or the grinding process used to make the crystalline samples. This needs to be included in the paper.
·'...shows axial signals at $g_{\text{perp}} = 2.3$, $g_{\parallel} = 2.0$ ($A//Co = 79.2$ G), which are typical...' would better read '...shows axial symmetry, with $g_{\text{perp}} = 2.3$, $g_{\parallel} = 2.0$ ($A//Co = 79.2$ G), which is typical...'
·You quote the characteristic resonances for the imidazolate compound as 6.0, 4.0 and 2.0, are these g values? If so they should be characterized as g_x , g_y and g_z . You then go onto give ' $A//Co = 82.0$ G' as it is usually expected that the g and A frames are coincident unless otherwise expressed, as such this notation is not correct here as g_x is not equal to g_y . The hyperfine values must be listed either in the same reference frame as the g tensor or if the reference frames are not the same the angles between them should be specified - these values should be easily extractable from your easyspin simulations. '...excited state and resonances at 4.0 and 2.0 come...' and '...weak signal is observed at ~ 2.3 ...' again are these g values? Please specify or give the plots in figure 2 on a g axis scale.

Thanks for your comments. Corrections and modifications have been given.

14. You state ‘In both phases a weak signal is observed at ~2.3 which might correspond to the four-coordinate [Co(TPP)]’ I do not agree with this. I would like to draw your attention to the paper: Probing the Surrounding of a Cobalt(II) Porphyrin and its Superoxo Complex by EPR Techniques by M. Baumgarten, C. J. Winscom, and W. Lubitz (Applied Magnetic Resonance February 2001, Volume 20, Issue 1-2, pp 35-70). In this work they study [Co(OEP)] in THF using ESR. They found that in this solvent the [Co(OEP)] could either remain 4 coordinate (base-off) or interact with the solvent to form a base-on state, both of which have very different ESR spectra (see figure 3 of the M. Baumgarten, C. J. Winscom, and W. Lubitz paper). In addition neither of these species were found to have a g value of 2.3 (See table 1 of the M. Baumgarten, C. J. Winscom, and W. Lubitz paper). The pure four coordinate spectrum has also been observed for Co(TPP) in a crystalline solid, and does not have g=2.3; as seen in Solid State Sciences Volume 9, Issues 3–4, March–April 2007, Pages 295-302 The characterization of cobalt(II) derivatives of selected substituted meso-tetraphenyl and tetrapyrrolyl porphyrins by EPR spectroscopic study by D. Skrzypek, I. Madejska and J. Havdas. I therefore do not believe that the signal you are observing at g=2.3 in the imidazolate samples is 4 coordinate [Co(TPP)]. In my mind it is more likely incompletely deprotonated sample and thus a residue signal from the imidazole compound. The reference spectra with 0 equivalents of imidazolate in THF is so badly saturated and therefore distorted (see my comment below about figure S6) that it is impossible in my opinion to use this as a comparison to the other datasets with larger equivalents of imidazolate.

Thanks for your comments and the references you gave.

1. The synthesis method of [K(2-MeIm⁻)] in principle excludes the incompletely deprotonated imidazole which would be washed out by the THF solvent. (according to Hu, C. *et al.* Just a Proton: Distinguishing the Two Electronic States of Five-Coordinate High-Spin Iron(II) Porphyrinates with Imidazole/ate Coordination. *J. Am. Chem. Soc.* **132**, 3737-3750 (2010)).

Synthesis of [K(2-MeIm⁻)]: [K(2-MeIm⁻)] was synthesized by reaction of 2-MeHIm with less than 1 equiv of KH in dry-box. Typically, 0.132 g of KH (3.30 mmol) was dissolved in 20 mL of THF, and 0.3 g of 2-MeHIm (3.66 mmol) was added over half hours. The resulting mixture was stirred 2 hours and the product was isolated by filtration, washed with THF (3×15 mL), and then dried under vacuum for 2 hours.

The method has been given in the SI.

We have also characterized [K(2-MeIm⁻)] and its starting material 2-MeHIm by HNMR, which are given in Figure S19 and S20. As can be seen, the NH proton has been entirely removed.

2. EPR spectra of kinds of different solution/solid samples have been re-measured and given in the text and SI. Carefully investigations on the data allow us to propose a low spin species i.e. weakly bonding intermediate [Co(TPP)⋯(2-MeIm⁻)]⁻ in the mother liquor which is responsible for the signals at 2.3. Please see the improved EPR part for detail.

It is worthy to note that the molecules in the mother liquor that accompanied isolated

UV-vis spectral change (in THF at 295 K) of 7.45×10^{-5} M solution of $[\text{Co}^{\text{II}}(\text{TPP})]$ upon addition of 0, 1, 3, 5, 10, 20 equiv. of $[\text{K}(222)(2\text{-MeIm}^-)]$.

crystals and showed EPR signals have been known for a while (Yatsunyk, L. A., Dawson, A., Carducci, M. D., Nichol, G. S. & Walker, F. A. Models of the Cytochromes: Crystal Structures and EPR Spectral Characterization of Low-Spin Bis-Imidazole Complexes of $(\text{OETPP})\text{Fe}^{\text{III}}$ Having Intermediate Ligand Plane Dihedral Angles. *Inorg. Chem.* **45**, 5417-5428 (2006)).

3. The strong signal of $\text{Co}(\text{TPP})$ at 2.3 in THF solution (without $[\text{K}(222)(2\text{-MeIm}^-)]$; black trace in the right panel of Figure S9) is also consistent with the idea of weakly bonding intermediate, because THF is known to weakly bond with metalloporphyrins (e.g. $[\text{Co}(\text{TPP}) \cdots \text{THF}]$) while PhCl almost does not (black trace in the left panel of Figure S9).
4. We have also monitored the UV-vis changes of the reaction which is given in the Figure. As can be seen, $\text{Co}(\text{II})$ is gradually oxidized to $\text{Co}(\text{III})$ upon the addition of $[\text{K}(222)(2\text{-MeIm}^-)]$ (Li, J. et al. Oxygenation of Cobalt Porphyrins: Coordination or Oxidation? *Inorg. Chem.* **49**, 2398-2406 (2010)). This and the always observed radical signals ($g = 2.0$) in both solution and solid samples of *imidazolate* derivatives (e.g. Figure S5, S6, S9-S12) suggest the addition of imidazolate has induced an oxidation which might be mediated by a radical e.g. $[\text{K}(222)^+]\cdot$. Such cation complexed crown ether radicals have been experimentally confirmed (e.g. $[\text{Na}(15\text{-C-5})^+]\cdot$ Rasmussen, J. K., Heilmann, S. M., Toren, P. E., Pocius, A. V. & Kotnour, T. A. Kinetics and mechanism of the interaction of potassium peroxydisulfate and 18-crown-6 in aqueous media. *J. Am. Chem. Soc.* **105**, 6845-6849 (1983); Kellner, I. D. et al. Ion formation pathways of crown ether–fullerene conjugates in the gas phase. *Phys. Chem. Chem. Phys.* **16**, 18982-18992 (2014)). Additional experimental and theoretical investigations on the oxidation mechanisms are still underway which will be reported in a separate paper.
5. It is worthy to note a radical signal ($g = 2.0$) is also observed for the solid $[\text{Co}(\text{TTP})(2\text{-MeHIm})]$ (Figure S8), while not for its solution sample (left bottom panel of Figure S12). We also found the radical signal appears stronger when the solid

sample staying longer in the well sealed EPR tube even under the dark. Hence, [Co(TTP)(2-MeHIm)] appears more unstable than [Co(TPP)(2-MeHIm)] counterpart which does not show any radical signal at both solid and solution states (Figure S7 & S11). Further studies on the stability of the two complexes are underway.

15. 'Hence, the high-spin resonances is assigned...' would read better 'Hence, the high-spin resonances are assigned...'

Thanks for your comments. Corrections have been made.

16. Figure S6: This figure is not suitable for publication - the spectra at 0 equivalents is clearly saturated - all of the signal is positive, whereas in continuous wave ESR the signal should be collected as a dispersive signal. It is likely that the problem could be relieved by increasing the temperature and reducing the microwave power used in the measurements. Unfortunately presentation of such data also makes me question the shape of the signals seen in the 3 and 7 equivalent spectra where the flat-tops of the low field feature look partially reminiscent truncated signals, I believe that this needs to be checked by the authors. All of the data in this figure needs to be re-measured at a higher temperature and a microwave power dependence study carried out for each set of conditions to ensure that the signals presented are not saturated.

Thanks for your comments. EPR spectra have been re-measured and given in text and SI.

Discussion:

17. I do not understand the final sentence, 'Given the complex protein environments of heme metal center, the current work implied negative character of proximal ligand does not readily make a complete spin state transition, while big enough metal displacements (i.e. $\Delta 4 > 0.2$ Angstrom) always involve with the d orbital reconstruction to perform the physiological functions.' Please reword.

Thanks for your comments. This paragraph has been modified.

Data availability:

18. Nature journals mandate the deposition of small molecule crystal structures in the Cambridge Structural Database, this should be done and the data availability statement corrected accordingly.

Thanks for your comments. Sorry about this, CCDC number are given and the data availability statement are corrected.

19. References:

The format of reference 10 is not correct.
36 is a footnote not a reference - please remove and put the information into the main text.

·41 is a footnote not a reference – please remove and put the information into the main text.

Thanks for your comments. These have been corrected and modified.

Reviewers' comments:

Reviewer #1 (Remarks to the Author):

In this revision the authors have improved the discussion and fixed many presentation issues (except some still remaining language problems, see examples below). I recommend the acceptance of this work after addressing the following points:

(1) As pointed out by Reviewer 3, in his/her comment #4, the abstract bears a lot of similarity to the abstract of ref 12 (previously ref 10). Specifically:

- The first sentence of abstract in ref 12: "A key step in cytochrome P450 catalysis includes the spin-state crossing from low spin to high spin upon substrate binding (...)"
- The first sentence of abstract in this work: "A key step in cytochrome P450 catalysis includes the spin state transition from low spin to high spin upon substrate addition."
- The last sentence of abstract in ref 12: "This is the first example of a synthetic iron(III) complex that can reversibly change its spin state between a high and an intermediate state through weak external perturbations."
- The last sentence of abstract in this work: "This is the first examples of synthetic metalloporphyrins that can switch the spin state through one proton of the proximal ligand."

It is clear and acceptable that related ideas are covered in these two papers. But so close similarity in the style is worrying (in terms of copyright) and completely unnecessary. The authors should rephrase these sentences using their own style.

(2) p. II. 1-8: I think the authors should mention, already here, that the observed structural features of the imidazolate complexes (large metal out-of-plane displacement, large Co-Np distances) are indicative of the HS state, which will be fully confirmed later by other techniques. The argument given in response to the comment #12 of Reviewer #2 is, in my opinion, correct and important, and it should be given also in the paper (to make it available to readers).

By the way, I found the discussion of single crystal structures quite challenging to follow (large number of symbols, abbreviations). Therefore, it would be nice - for the clarity and soundness of the presentation - to summarize this part with a clear conjecture of the HS state for the imidazolate complexes based on the crystal structures.

(3) Concerning comment #4 of Reviewer 2, I think the authors should mention Figure S3 in the footnote h of Table (the footnote defining angle PHI). This would help readers to follow the discussion.

(4) p.5, l.10 "Shoestring" diagrams - this term should be also used literarily in the caption of figure S4 in SI.

Also, concerning caption of figure s4:

- "diagrams illustrate" should be perhaps "diagrams illustrating"
- "All diagrams are started at C(m1 of the porphyrin atoms." - unclear

(4) Some style & language issues:

- p.3, l. 14 mössbauer (should be capitalized)
- p.4, l.8 "are high-spin state" should be changed perhaps to: "are high-spin" or "have high-spin ground states".
- p.4, l.10 the short customary name "cryptand 222" or "[2.2.2]cryptand" should be preferred over the systematic IUPAC name for this well-known species (this is a suggestion only).
- p. 5, l.13 "the strong steric repulsion between imidazole plane and Co-Np vector" . Repulsion between "plane" and "vector" is a jargon, which should be avoided.
- p. 11, ll.2-4 "The dramatically lowered dxz, dyz, dz2 and dx2-y2 orbitals of HS state is obvious

which is consistent with the short axial bond distance and large metal displacement (Table 1)."
Possible grammatical issues with this sentence.

- p.12, ll.4-6 "Although both are mediated by N–H proton of imidazole, the spin state transition happened only to cobalt(II) porphyrinates, in contrast to invariable high-spin states of 2-methylimidazole(ate) iron(II) analogues." To which object "mediated" apply in this sentence?
- p. 13, l. 14 "negative character changes of proximal ligands" This phrase sounds strange.

My general impression is that the writing of this paper could be still improved, in order to make it more attractive to the readers.

Reviewer #3 (Remarks to the Author):

While the authors have addressed many of my concerns raised in my original review, for which I thank them, there are still problems with some of the EPR data presented in the SI and the interpretation of this data. Specifically there are discrepancies in terms of parameter values given in both the SI and the main text which make understanding and interpreting the data in the paper as a reader impossible. There are two significantly different EPR data sets which appear to have been recorded on the same compound prepared separately with no explanation offered as to why the differences arise. The EPR simulations in the main text do not match those given in the SI and there is still a saturated EPR spectrum in the SI which should not be published and needs to be re-recorded under conditions where the data is not saturated.

My concerns in more detail are:

1) Comparing Figure 2 in the main paper to figure S11 in the SI, I assume that the crystalline spectra presented for both these figures are the same? Although no mention of the crystalline spectra is given in the figure caption of S11. If this is the case why are the simulations for the [Co(TPP)(2-MeIm-)]- species clearly different? Only one set of simulation parameters for this species is given in the SI?

2) Figure S5. From the wording of the figure caption I do not understand the difference between the blue trace in the figure on the left and the black trace in the figure on the right other than they were the same sample prepared twice at different times and measured separately. If this is the case it is worrying that the signals measured are so very different. No explanation of the differences is given and this should be addressed.

3) In the figure caption to figure S5 the author's state: 'The signal at 2.0 is supposed to be an radical which is generated upon the addition of imidazolate. The mechanism studies are underway which will be reported in a separate paper.' In their response to the reviewers they state that they believe that this signal is due to oxygenated Cobalt(III) porphyrin and provide evidence for this in the form of UV-vis spectra. This would make sense and the authors should include this information in the paper.

4) Figure S8. The data presented for Co(TPP) in THF with 0 equivalents of [K(222)(2-MeIm-)] black trace in figure on the right is still badly saturated in spite of the fact that the authors say they have re-measured this data set. Data measured from normal methods of continuous wave EPR must be dispersive showing both a positive and a negative part. The distorted line shape suggests that the spectrum was measured using too high a microwave power for the conditions used. This could be remedied by re-measuring the data at a significantly higher temperature, lower power or combination of both. For example the spectrum of the related compound Co(II)OEP in THF in Figure 3 of Probing the Surrounding of a Cobalt(II) Porphyrin and its Superoxo Complex by EPR Techniques by M. Baumgarten, C. J. Winscom, and W. Lubitz (Applied Magnetic Resonance February 2001, Volume 20, Issue 1-2, pp 35-70) was measured at 140K. The spectrum presented in that paper is free of saturation effects. Saturation effects in CW EPR and the resulting distortion of line shapes are described in Jeschke G. (2007) Instrumentation and Experimental Setup. In:

ESR Spectroscopy in Membrane Biophysics. Biological Magnetic Resonance, vol 27. Springer, Boston, MA. See section 1.4. Setting Receiver Gain and m.w. Power.

5) The figure captions for S11 and S12 do not describe the crystalline spectra presented. These should be changed to reflect the data shown in the figure.

6) On page 6 lines 20-21 of the main paper the authors state 'As can be seen, the two imidazole derivatives ([Co(TPP)(2-MeHIm)] and [Co(TTP)(2-MeHIm)]) show consistent axial symmetric spectra with $g_{\perp} = 2.3$, $g_{\parallel} = 2.0$ ($A//Co = 79.2$ G),' however this A value do not agree with those given as the simulation parameters in the SI '[Co(TPP)(2-MeHIm)] Simulation parameters: $S = 1/2$; $g = [2.3, 2.0]$; $A(59Co) = -[5, 250]$ MHz; $lwpp = 6$ linewidth mT.' Converting the precise value of $A//Co = 79.2$ G given in the main paper into MHz gives a value of 222.0 MHz (3 s.f.) not the 250 MHz given in the SI. This discrepancy needs to be rectified so that a correct and consistent set of values are used.

7) Likewise a value of ($A//Co = 82.0$ G) is given for the ([Co(TPP)(2-MeIm-)] – species, which while in better agreement with the value of 232.4 MHz given in the SI is not completely in agreement as a value of 232.4 MHz would correspond to a value of 82.9G (3 s.f.).

8) On page 7 of the main text, lines 5-6, the authors state 'The zero value of E/D yielded by simulations confirmed the axial system' However in the SI the calculation parameters shown give an E/D value of $E/D = 0.125$ which is non-zero and would indicate a departure from axial symmetry towards a rhombic system (which would have $E/D = 1/3$). The main text about features arising from different Kramer's doublets is only valid for an axial system with $E/D = 0$.

9) The manuscript and SI contain several typos (for example 'smapple ') and needs careful proofreading.

Reviewer 1

1. As pointed out by Reviewer 3, in his/her comment #4, the abstract bears a lot of similarity to the abstract of ref 12 (previously ref 10). Specifically:

- The first sentence of abstract in ref 12: "A key step in cytochrome P450 catalysis includes the spin-state crossing from low spin to high spin upon substrate binding (...)"
- The first sentence of abstract in this work: "A key step in cytochrome P450 catalysis includes the spin state transition from low spin to high spin upon substrate addition."

- The last sentence of abstract in ref 12: "This is the first example of a synthetic iron(III) complex that can reversibly change its spin state between a high and an intermediate state through weak external perturbations."
- The last sentence of abstract in this work: "This is the first examples of synthetic metalloporphyrins that can switch the spin state through one proton of the proximal ligand."

It is clear and acceptable that related ideas are covered in these two papers. But so close similarity in the style is worrying (in terms of copyright) and completely unnecessary. The authors should rephrase these sentences using their own style.

Thanks for your comments. Modifications have been made on the two sentences.

2. p. II. 1-8: I think the authors should mention, already here, that the observed structural features of the imidazolate complexes (large metal out-of-plane displacement, large Co-Np distances) are indicative of the HS state, which will be fully confirmed later by other techniques. The argument given in response to the comment #12 of Reviewer #2 is, in my opinion, correct and important, and it should be given also in the paper (to make it available to readers).
By the way, I found the discussion of single crystal structures quite challenging to follow (large number of symbols, abbreviations). Therefore, it would be nice - for the clarity and soundness of the presentation - to summarize this part with a clear conjecture of the HS state for the imidazolate complexes based on the crystal structures.

Thanks for your comments. Modifications have been made in the crystal structure part.

3. Concerning comment #4 of Reviewer 2, I think the authors should mention Figure S3 in the footnote h of Table (the footnote defining angle PHI). This would help readers to follow the discussion.

Thanks for your comments. This has been given.

4. p.5, 1.10 "Shoestring" diagrams - this term should be also used literarily in the caption of figure

S4 in SI. Also, concerning caption of figure s4:

- "diagrams illustrate" should be perhaps "diagrams illustrating"
- "All diagrams are started at C(m1 of the porphyrin atoms." - unclear

Thanks for your comments. Corrections and modifications have been made in the SI.

Some style & language issues:

- p.3, l. 14 mössbauer (should be capitalized)

Thanks for your comments. This has been corrected.

- p.4, l.8 "are high-spin state" should be changed perhaps to: "are high-spin" or "have high-spin ground states".

Thanks for your comments. This has been modified.

- p.4, l.10 the short customary name "cryptand 222" or "[2.2.2]cryptand" should be preferred over the systematic IUPAC name for this well-known species (this is a suggestion only).

Thanks for your comments. This is the first time "222" is mentioned in the text, so it is formally defined.

- p. 5, l.13 "the strong steric repulsion between imidazole plane and Co–Np vector" . Repulsion between "plane" and "vector" is a jargon, which should be avoided.

Thanks for your comments. This has been modified.

- p. 11, ll.2-4 "The dramatically lowered dxz, dyz, dz2 and dx2–y2 orbitals of HS state is obvious which is consistent with the short axial bond distance and large metal displacement (Table 1)." Possible grammatical issues with this sentence.

Thanks for your comments. "The dramatically lowered dxz, dyz, dz2 and dx2–y2 orbitals of HS state is obvious which is consistent with the short axial bond distance and large metal displacement (Table 1)." has been modified to : "The dramatically lowered dxz, dyz, dz2 and dx2–y2 orbitals of HS state which is consistent with the short axial bond distance and large metal displacement is obvious (Table 1)"

- p.12, ll.4-6 "Although both are mediated by N–H proton of imidazole, the spin state transition happened only to cobalt(II) porphyrinates, in contrast to invariable high-spin states of 2-methylimidazole(ate) iron(II) analogues." To which object "mediated" apply in this sentence?

Thanks for your comments. "Although both are mediated by N–H proton of imidazole, the spin state transition happened only to cobalt(II) porphyrinates, in contrast to invariable high-spin states of 2-methylimidazole(ate) iron(II) analogues." has been modified to: "The spin state

transition happened only to cobalt(II) porphyrinates, in contrast to invariable high-spin states of 2-methylimidazole(ate) iron(II) analogues, although both pairs of counterparts are mediated by N–H proton of imidazole.”

- p. 13, l. 14 "negative character changes of proximal ligands" This phrase sounds strange. My general impression is that the writing of this paper could be still improved, in order to make it more attractive to the readers.

Thanks for your comments. "negative character changes of proximal ligands" has been modified to : charge changes of proximal ligands.

Reviewer 3

1. Comparing Figure 2 in the main paper to figure S11 in the SI, I assume that the crystalline spectra presented for both these figures are the same? Although no mention of the crystalline spectra is given in the figure caption of S11. If this is the case why are the simulations for the [Co(TPP)(2-MeIm–)]– species clearly different? Only one set of simulation parameters for this species is given in the SI?

We are sorry about the mistakes. The crystalline spectra presented in Figure 2 and Figure S11 are same. The simulations have been redone and given in the figures and SI.

2. Figure S5. From the wording of the figure caption I do not understand the difference between the blue trace in the figure on the left and the black trace in the figure on the right other than they were the same sample prepared twice at different times and measured separately. If this is the case it is worrying that the signals measured are so very different. No explanation of the differences is given and this should be addressed.

Thanks for your comments. Modifications have been done and given in the caption of Figure S5.

3. In the figure caption to figure S5 the author’s state: ‘The signal at 2.0 is supposed to be an radical which is generated upon the addition of imidazolate. The mechanism studies are underway which will be reported in a separate paper.’ In their response to the reviewers they state that they believe that this signal is due to oxygenated Cobalt(iii) porphyrin and provide evidence for this in the form of UV-vis spectra. This would make sense and the authors should include this information in the paper.

Thanks for your comments. The UV-vis spectra are given in Figure S19 and modifications are made in the caption of Figure S5.

4. Figure S8. The data presented for Co(TPP) in THF with 0 equivalents of [K(222)(2-MeIm–)] black trace in figure on the right is still badly saturated in spite of the fact that the authors say

they have re-measured this data set. Data measured from normal methods of continuous wave EPR must be dispersive showing both a positive and a negative part. The distorted line shape suggests that the spectrum was measured using too high a microwave power for the conditions used. This could be remedied by re-measuring the data at a significantly higher temperature, lower power or combination of both. For example the spectrum of the related compound Co(II)OEP in THF in Figure 3 of Probing the Surrounding of a Cobalt(II) Porphyrin and its Superoxo Complex by EPR Techniques by M. Baumgarten, C. J. Winscom, and W. Lubitz (Applied Magnetic Resonance February 2001, Volume 20, Issue 1-2, pp 35-70) was measured at 140K. The spectrum presented in that paper is free of saturation effects. Saturation effects in CW EPR and the resulting distortion of line shapes are described in Jeschke G. (2007) Instrumentation and Experimental Setup. In: ESR Spectroscopy in Membrane Biophysics. Biological Magnetic Resonance, vol 27. Springer, Boston, MA. See section 1.4. Setting Receiver Gain and m.w. Power.

Thanks for your comments and the reference papers.

EPR spectra of Co(TPP) in THF with 0 equivalent of [K(222)(2-MeIm⁻)] has been re-measured and given in Figure S9.

The paper (Applied Magnetic Resonance February 2001, Volume 20, Issue 1-2, pp 35-70) which gave elegant studies on Co(II)OEP in THF has been cited in both text and SI.

5. The figure captions for S11 and S12 do not describe the crystalline spectra presented. These should be changed to reflect the data shown in the figure.

Thanks for your comments. These have been modified.

On page 6 lines 20-21 of the main paper the authors state ‘As can be seen, the two imidazole derivatives ([Co(TPP)(2-MeHIm)] and [Co(TTP)(2-MeHIm)]) show consistent axial symmetric spectra with $g_{\perp} = 2.3$, $g_{\parallel} = 2.0$ ($A//Co = 79.2$ G),’ however this A value do not agree with those given as the simulation parameters in the SI ‘[Co(TPP)(2-MeHIm)] Simulation parameters: $S = 1/2$; $g = [2.3, 2.0]$; $A(59Co) = -[5, 250]$ MHz; $lwpp = 6$ linewidth mT.’ Converting the precise value of $A//Co = 79.2$ G given in the main paper into MHz gives a value of 222.0 MHz (3 s.f.) not the 250 MHz given in the SI. This discrepancy needs to be rectified so that a correct and consistent set of values are used.

Likewise a value of ($A//Co = 82.0$ G) is given for the ([Co(TPP)(2-MeIm⁻)] – species, which while in better agreement with the value of 232.4 MHz given in the SI is not completely in agreement as a value of 232.4 MHz would correspond to a value of 82.9G (3 s.f.).

We are sorry about the mistakes. The simulations have been redone.

For [Co(TPP)(2-MeHIm)], $A_{//}^{Co} = 79.2$ (experimental) and 79 G (simulation).

For [Co(TPP)(2-MeIm⁻)], $A_{//}^{Co} = 82$ (experimental) and 82 G (simulation).

On page 7 of the main text, lines 5-6, the authors state ‘The zero value of E/D yielded by simulations confirmed the axial system’ However in the SI the calculation parameters shown

give an E/D value of $E/D = 0.125$ which is non-zero and would indicate a departure from axial symmetry towards a rhombic system (which would have $E/D = 1/3$). The main text about features arising from different Kramer's doublets is only valid for an axial system with $E/D = 0$.

We are sorry about the mistakes. The simulations have been redone. The new parameters give $E/D = 0$, in consistent with the crystal structure and DFT calculations.

6. The manuscript and SI contain several typos (for example 'smapple ') and needs careful proofreading.

Thanks for your comments. Corrections and modifications have been made.

REVIEWERS' COMMENTS:

Reviewer #3 (Remarks to the Author):

Referring to numbering provided in my previous review.

1) The corrections to figures 2 and S11 are ok.

2) In figure S5 while the preparation conditions for the two different samples are now given (grinding in a quartz pestle and mortar vs an agate one). No explanation is offered as to why the spectra are so very different. I believe that this still needs to be addressed and an explanation offered as to why the different grinding methods result in different spectra and the species to which these spectra belong.

3) In their haste to correct the figures in the supporting information the authors have omitted the asterisks from figures S5, S6, S9 and S10 referred to on line 12 page 7 which mark the signal at 2.0. this needs to be corrected as the asterisk notation is referred to in the main paper.

4) The correction to figure S9 is fine as are the corrections to the simulations.

Points 2 and 3 on the above list require addressing and correcting before publication.

REVIEWERS' COMMENTS:

Reviewer #3 (Remarks to the Author):

Referring to numbering provided in my previous review.

1) The corrections to figures 2 and S11 are ok.

Thanks!

2) In figure S5 while the preparation conditions for the two different samples are now given (grinding in a quartz pestle and mortar vs an agate one). No explanation is offered as to why the spectra are so very different. I believe that this still needs to be addressed and an explanation offered as to why the different grinding methods result in different spectra and the species to which these spectra belong.

Thanks for your comments. Here is the modified legend. Multi-temperature EPR spectra of crystalline $[\text{Co}(\text{TPP})(2\text{-MeIm}^-)]^-$. a: the sample was grinded inside the EPR tube by a quartz pestle. Signals at 6.0, 4.0, 2.0 belong to high-spin $[\text{Co}(\text{TPP})(2\text{-MeIm}^-)]^-$. The signal at 2.3 is supposed to be $[\text{Co}(\text{TPP})\cdots(2\text{-MeIm}^-)]^-$ intermediate which was generated during reaction and accompanied the isolated crystals through the mother liquor and/or was generated during grinding process. b: the sample was grinded by the agate mortar. The signal at 5.5 belongs to high-spin $[\text{Co}(\text{TPP})(2\text{-MeIm}^-)]^-$. The strong signal at 2.3 suggests vigorous grinding has produced more $[\text{Co}(\text{TPP})\cdots(2\text{-MeIm}^-)]^-$ intermediate. The signal at 2.0 is supposed to be a radical which is generated during the oxidation of Co(II) to Co(III) upon addition of imidazolate (Supplementary Figure 19). The mechanism studies are underway which will be reported in a separate paper.

3) In their haste to correct the figures in the supporting information the authors have omitted the asterisks from figures S5, S6, S9 and S10 referred to on line 12 page 7 which mark the signal at 2.0. this needs to be corrected as the asterisk notation is referred to in the main paper.

Thanks for your comments. These have been modified.

4) The correction to figure S9 is fine as are the corrections to the simulations.

Points 2 and 3 on the above list require addressing and correcting before publication.

Thanks!